# **SURGE:** Approximation and Training Free Particle Filter for Diffusion Surrogate

**Lifu Wei** [1]   **Yinuo Ren** [2]   **Naichen Shi** [3 1]   **Yiping Lu** [3]

## Abstract

Data assimilation (DA) tackles the sequential estimation of a dynamical system's latent state from noisy, partial observations. In this paper, we study DA in a setting where the system dynamics are represented by a pretrained diffusion model used as a surrogate forecaster. We focus on how to integrate incoming observations into the diffusion surrogate's predictions to support continuous state correction and progressively refining the estimated trajectory over time. After receiving noisy observations, the diffusion model is guided using the observation likelihood to steer the generation process toward observation-consistent states. However, such guidance does not guarantee sampling from the true posterior. Motivated by particle filtering methods, we represent the posterior distribution using an ensemble of particles. We perform Sequential Monte Carlo over diffusion trajectories, working with their path measure. We compute importance weights for generated particles and resample to focus on trajectories consistent with the observations. This procedure corrects the generation dynamics, drives the particle approximation toward the desired posterior and leads to an approximation-free particle filtering method that rigorously fuses observational data with diffusion model simulations.

## 1. Introduction

Recovering the state of complex dynamical systems from noisy and incomplete observations is a fundamental problem in science and engineering. Accurate state estimation

---
[1]Department of Mechanical Engineering, Northwestern University, Evanston, IL, United States [2]Institute for Computational & Mathematical Engineering, Stanford University, Stanford, CA, United States [3]Department of Industrial Engineering & Management Sciences, Northwestern University, Evanston, IL, United States. Correspondence to: Yiping Lu <yiping.lu@northwestern.edu>.

*Proceedings of the 43rd International Conference on Machine Learning*, Seoul, South Korea. PMLR 306, 2026. Copyright 2026 by the author(s).

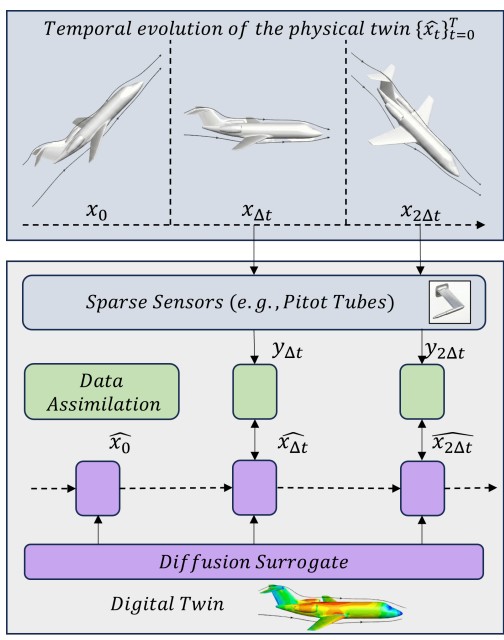

*Figure 1.* Many modern systems come with complementary "digital" and "physical" twins: a diffusion-based digital twin learns from historical trajectories to generate probabilistic forecasts over dynamics, while sparse and noisy measurements from the physical twin are assimilated to correct and fuse multiple predictions into a single posterior state estimate.

underpins reliable prediction in applications such as weather forecasting, oceanography, seismology, and fluid dynamics, where system evolution is governed by nonlinear, high-dimensional, and often stochastic partial differential equations. Data assimilation (DA) addresses this challenge by combining model-based forecasts with observational data to produce physically consistent state estimates.

In many modern settings, we increasingly have access to two complementary "twins" of the same system (Reich, 2025). On one hand, a data-driven digital twin can be learned from historical trajectories, for which diffusion models are a particularly suitable choice because they can represent high-dimensional complex distributions and generate realistic stochastic evolutions (Ho et al., 2022; Gruver et al., 2023; Li et al., 2022; Nie et al., 2025). On the other hand, a physical twin delivers measurements from the real system, often sparse, noisy, and incomplete, but grounded in reality. This motivates a DA perspective in which a diffusion-based digital twin produces probabilistic forecasts, while observations from the physical twin are used to correct and aggregate

multiple predictive sources into a single posterior estimate of the system state.

However, when the underlying dynamics are simulated by a diffusion model rather than an explicit mechanistic time-stepping solver, it is nontrivial to design a filtering/assimilation framework that incorporates physical observations in a principled way. In particular, a key challenge is how to fuse the diffusion model's prior forecast distribution with the physical twin's observations to obtain an approximation-free (or controlled-bias), physically consistent posterior, while remaining computationally tractable in the high-dimensional, nonlinear regime.

Mathematically, a discrete-time stochastic dynamical system can be described by:

$$x_{t+1} = \Psi(x_t, \xi_t), \qquad y_{t+1} = \mathcal{A}(x_{t+1}) + \varepsilon_{t+1}, \quad (1)$$

wher $x_t \in \mathbb{R}^D$ is the state vector at time step $k$, $\Psi$ is the state transition map and $\xi_t$ denotes the stochastic force. The observations $y_t \in \mathbb{R}^M$ are related to the state through the measurement map $\mathcal{A}$, with observation noise $\varepsilon_{t+1}$. DA seeks to infer the posterior of the state trajectory $x_{0:T}$ given observations $y_{1:T}$.

This paper focuses on the case where a dynamic system is simulated using a generative digital twin. More specifically, the conditional transition density $p(x_{t+1} \mid x_t)$ of dynamical system with transition $x_{t+1} = \Psi(x_t, \xi_t)$ is approximated using a conditional diffusion model (Song et al., 2020; Lipman et al., 2022; Liu et al., 2022; Albergo & Vanden-Eijnden, 2022; Albergo et al., 2025). Sampling the next state is implemented by simulating an internal diffusion time process that maps a base distrbution to a sample in state space. When a new observation arrives, a common practice is to guide diffusion sampling by the observation likelihood, swterring samples toward observation-consistent states. However, unless the guidance is the exact Doob's $h$-transform (Denker et al., 2024; Tang & Xu, 2024; Nguyen et al., 2025; Ren et al., 2025b; Sabour et al., 2025; Sun, 2025; Domingo-Enrich et al., 2024; Havens et al., 2025; Liu et al., 2025; Zhu et al., 2026), guided sampling does not in general produce samples from the correct posterior. Exact Doob guidance typically requires solving an associated backward Kolmogorov equation (Albergo & Vanden-Eijnden, 2024; Zhang et al., 2023), which is intractable in high-dimensional generative models. In most cases, the resulting guidance is necessarily approximate and often heuristic (Ren et al., 2025a; Chen et al., 2025b). A natural question is thus:

**How can we fuse diffusion-based digital twin simulations with partial observations without introducing systematic bias from approximate guidance?**

To construct an approximation-free data assimilation frame-

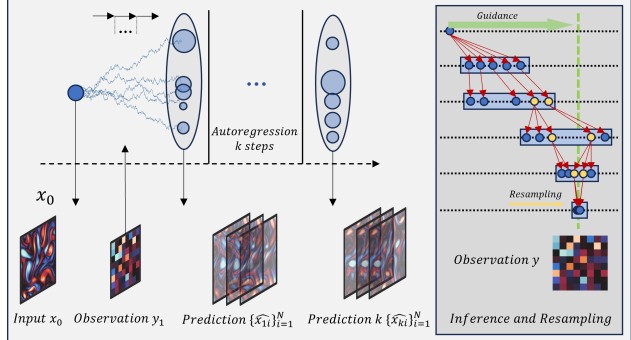

*Figure 2.* Conceptual description of SURGE for Data Assimilation. We present SURGE, an approximation-free data assimilation framework. SURGE presents diffusion-based process as a path distribution and perform reweighting and resampling on multi stochastic trajectories during inference to achieve approximation-free output.

work, we draw inspirations from particle filtering, which approximates the filtering posterior by propagating and reweighting an ensemble of particles according to the system dynamics and the observation likelihood. In our approach, we evaluate the probability of each generated trajectory using the Girsanov change of measure, which accounts for the discrepancy between the guided diffusion model dynamics and the target dynamics. We perform Sequential Monte Carlo (SMC) updates at each step to reweight and resample particles. This sequential resampling mitigates particle degeneracy and ensures that the particle ensemble remains concentrated in regions of high posterior probability. To reduce resampling variance, we incorporate the observation likelihood progressively along the trajectory, so that particle weights remain well-balanced at each small time step rather than becoming highly concentrated due to a single, large likelihood update. Building on this idea, we propose the SURGE filter (Sequential Unbiased Resampling via Girsanov Estimation), a particle filtering-based framework that enables approximation-free data assimilation for diffusion-model-based simulators. Operating exclusively at the inference phase, SURGE is designed to enhance performance in high-dimensional, long-horizon physical processes characterized by sparse and noisy observations.

**Contributions.** Our contributions are as follows:

- We introduce SURGE, an approximation-free data assimilation method that rigorously combines a diffusion surrogate-based digital twin with noisy, partial measurements from a physical twin via particle filtering, enabling sequential state correction over time.
- We cast diffusion-based forecasting as inference over a path distribution and perform reweighting and resampling on diffusion trajectories, using a Girsanov change-of-measure to compute importance weights that correct bias introduced by likelihood guidance. We further stabilize filtering in high dimensions via progressive (time-

distributed) likelihood incorporation, mitigating particle degeneracy and rigorously fusing observations with diffusion-model simulations.

- We apply SURGE solely at the inference level of prevailing diffusion-based data assimilation methods, evaluating it under conditions of long-time sequence prediction and high-dimensional chaotic dynamics. The findings confirm that SURGE adeptly governs sequential generation for diffusion surrogates, enhancing performance during the inference phase.

## 2. Preliminaries

In this section, we will briefly review the background of data assimilation and particle filtering.

### 2.1. Data Assimilation

Given the stochastic dynamical system (1), the goal of data assimilation (Sanz-Alonso et al., 2023) is to approximate the posterior distribution

$$p(x_{0:T} \mid y_{1:T}) \propto p(x_0) \prod_{t=0}^{T-1} p(x_{t+1} \mid x_t) p(y_{t+1} \mid x_{t+1}).$$

This posterior can be sequentially approximated via a prediction step and an analysis step.

**Prediction Step.** In the prediction step, the current posterior is propagated forward through the dynamical model to obtain a prior distribution. Suppose the dynamical system (1) has the transition kernel $p(x_{t+1} \mid x_t)$, the prediction step gives the forward predict of latent state at time $t + 1$ as

$$p(x_{t+1} \mid y_{1:t}) = \int p(x_{t+1} \mid x_t) p(\mathrm{d}x_t \mid y_{1:t}). \quad (2)$$

**Analysis Step.** In the analysis step, the prior is updated using incoming observations to produce the posterior at the current time. At time $t + 1$, it updates the predictive prior with the new observation $y_{t+1}$ through the likelihood:

$$p(x_{t+1} \mid y_{1:t+1}) \propto p(y_{t+1} \mid x_{t+1}) p(x_{t+1} \mid y_{1:t}).$$

By alternating between the prediction and analysis steps, the posterior evolves from $p(x_t \mid y_{1:t})$ to $p(x_{t+1} \mid y_{1:t+1})$.

### 2.2. Particle Filters

Within the above discrete-time stochastic dynamical system, particle filtering (Gordon et al., 1993; Kitagawa, 1996; Del Moral, 1996; Johansen, 2009) provides a sampling-based approach to approximate the posterior distribution of the system state given sequential observations. The key idea is to represent the filtering distribution by a collection of weighted particles that evolve according to the system dynamics and are corrected using incoming data.

Assume that at time $t$, the posterior distribution $p(x_t \mid y_{1:t})$ is approximated by an ensemble of $M$ particles

$$p(x_t \mid y_{1:t}) \approx \frac{1}{M} \sum_{m=1}^{M} \delta_{x_t^{(m)}}(x_t),$$

where $\{x_t^{(m)}\}_{m=1}^M \subset \mathbb{R}^D$ are $M$ particle with equal weights. The particle filter proceeds recursively from time $t$ to $t + 1$ through the following three steps.

**Prediction Step.** Each particle is propagated forward using the stochastic state transition map $\Psi$ (1). Specifically, for each $x_t^{(m)}$, a predicted particle is generated by $\hat{x}_{t+1}^{(m)} = \Psi(x_t^{(m)}, \xi_t^{(m)})$, where $\xi_t^{(m)}$ is an independent realization of $\xi_t$. The collection of predicted particles $\{\hat{x}_{t+1}^{(m)}\}_{m=1}^M$ defines an empirical approximation of the prior distribution $p(x_{t+1} \mid y_{1:t}) \approx \frac{1}{M} \sum_{m=1}^M \delta_{\hat{x}_{t+1}^{(m)}}(x_{t+1})$.

**Update Step.** With the new observation $y_{t+1}$, the predicted particle $\hat{x}_{t+1}^{(m)}$ are reweighted according to the importance weight given by $w_{t+1}^{(m)} \propto p(y_{t+1} \mid x_{t+1} = \hat{x}_{t+1}^{(m)})$. This yields a weighted empirical approximation of the posterior $p(x_{t+1} \mid y_{1:t+1}) \approx \sum_{m=1}^M w_{t+1}^{(m)} \delta_{\hat{x}_{t+1}^{(m)}}(x_{t+1})$.

**Resampling Step.** To mitigate particle degeneracy, resampling is applied to replace the weighted ensemble with an equally weighted set of particles $\{x_{t+1}^{(m)}\}_{m=1}^M$ drawn from the above weighted distribution. The resulting empirical measure $p(x_{t+1} \mid y_{1:t+1}) \approx \frac{1}{M} \sum_{m=1}^M \delta_{x_{t+1}^{(m)}}(x_{t+1})$ serves as the filtering distribution at time $t + 1$, completing one assimilation cycle.

Despite its conceptual generality, particle filtering is known to suffer from severe weight degeneracy in high-dimensional state spaces, where the number of particles required to avoid collapse grows exponentially with the system dimension, rendering the method numerically less efficient and scalable in practical data assimilation settings.

### 2.3. Related Works

**Diffusion Model for Data Assimilation** In diffusion-based data assimilation, the reverse-time sampler requires the score of the diffused posterior, $\nabla_{x_t} \log p_t(x_t \mid y_{1:t})$. However, $p_t(x_t \mid y_{1:t})$ is the marginal obtained by diffusing the true posterior $p(x_t \mid y_{1:t})$ through the forward noising kernel, i.e., $p_t(x_t \mid y) = \int p_t(x_t \mid x_0) p(x_0 \mid y) \mathrm{d}x_0$. This marginalization makes the posterior score analytically intractable in general. Consequently, practical methods must approximate this term (Rozet & Louppe, 2023; Bao et al., 2023; Hodyss & Morzfeld, 2025). Another line of research leverages guidance for flow matching (Chen et al., 2025b; Transue et al., 2025; Bao & Sun, 2025) to construct filters for diffusion surrogates, so that generated samples are consistent with partial observations. However, these

guidance procedures remain approximate: they do not in general target the exact posterior, and they cannot provide approximation-free samples from the reward-tilted distribution. To the best of the authors' knowledge, this is the first work to provide an approximation-free filtering algorithm for diffusion-based surrogates of the posterior distribution.

**Inference-Time Scaling with Diffusion Prior** When a diffusion model is used as a learned prior, downstream inference can be formulated as approximate posterior sampling (Xu & Chi, 2024; Wu et al., 2024; Coeurdoux et al., 2024; Bruna & Han, 2024) or optimization under measurement or task constraints, where each denoising step refines the estimate while enforcing consistency (e.g., via gradient-based corrections or guidance). Many previous works (Wu et al., 2023; Cardoso et al., 2023; Singhal et al., 2025; Chen et al., 2025a; Ren et al., 2025a; Skreta et al., 2025; Uehara et al., 2025; He et al., 2026; 2025; Ou et al., 2025; Chu et al., 2025; Hasan et al., 2026; Ma et al., 2025) explored the use of particle filters or related methods for better posterior approximation and constraint satisfaction, making diffusion priors a natural substrate for test-time compute scaling without retraining the prior.

**Nonlinear Filtering for Diffusions** Nonlinear filtering for diffusion processes concerns inference of a latent continuous-time state $X_t$ evolving as an SDE from partial and noisy observations $Y_t$. The optimal conditional law $\pi_t = \mathcal{L}(X_t \mid \mathcal{F}_t^Y)$ is characterized by the Kushner–Stratonovich equation (Kushner, 1967; Stratonovich, 1965), with an equivalent unnormalized Zakai SPDE (Zakai, 1969). While these SPDE characterizations are complete, they are typically intractable beyond special cases, especially in high dimensions. Sequential Monte Carlo (SMC) / particle filtering methods (Sottinen & Särkkä, 2008; Fearnhead et al., 2008; Del Moral et al., 2006; Doucet et al., 2000; Wang et al., 2026; Zhu & Lu, 2026), including continuous-time variants, are therefore widely used, but often suffer from weight degeneracy and poor scaling as dimension grows.

## 3. SURGE: Sequential Unbiased Resampling via Girsanov Estimation

We aim to perform data assimilation by combining physical observations with simulations from a digital twin which, in our setting, is implemented as a conditional diffusion model approximating the system dynamics; prior works (Chen et al., 2025b; Transue et al., 2025) rely on guidance mechanisms, as they enable observational constraints to be injected into the sampling process without retraining or explicitly modifying the generative model, leading to stable and effective posterior correction.

### 3.1. Conditional Diffusion Surrogate for Transitions

Starting from samples drawn from $p(x_t \mid y_{1:t})$, we assume that the one-step transition $p(x_{t+1} \mid x_t)$ is modeled using a *conditional diffusion* sampler (Lipman et al., 2022; Tong et al., 2023). It can be expressed as an stochastic differential equation (SDE) on an internal time variable $s \in [t, t + 1]$:

$$\mathrm{d}x_s = v_\theta(x_s, s - t \mid x_t)\mathrm{d}s + \Sigma^{1/2}(s - t)\mathrm{d}W_s, \quad (3)$$

which defines a path measure, denoted as $\mathbb{P}(x_{t:t+1}|y_{1:t})$, on the space of continuous trajectories $C([t, t + 1], \mathbb{R}^D)$, with marginals $\mathbb{P}(x_t \mid y_{1:t}) = p(x_t \mid y_{1:t})$ and $\mathbb{P}(x_{t+1} \mid y_{1:t}) = p(x_{t+1} \mid y_{1:t})$.

To encourage consistency between the generated next state and the observations, one often modifies the drift during sampling using a guidance term derived from the likelihood (Nichol et al., 2021; Ho & Salimans, 2022). Given an observation $y_{t+1}$, a guided sampler takes the form:

$$\begin{aligned}\mathrm{d}x_s^G = \big[&v_\theta(x_s^G, s - t \mid x_t) \\ &+ \Sigma(s - t)\nabla_x G(x_s^G, s - t \mid y_{t+1})\big]\mathrm{d}s + \Sigma^{1/2}(s - t)\mathrm{d}W_s,\end{aligned}$$
$$(4)$$

where $G : \mathbb{R}^d \times [0, 1] \to \mathbb{R}$ is a control potential, *e.g.*, guidance from a prior model or simplified physics. Similarly, we denote the path measure of the SDE (4) as $\mathbb{P}^G(x_{t:t+1}^G \mid y_{1:t+1})$.

When $G$ equals $\log h$ for the correct Doob's $h$-transform, the guided process corresponds to the exact conditioned dynamics. In practice, a common choice is to set

$$\Sigma(s - t)\nabla_x G(x_s, s \mid y_{t+1}) = \lambda(s - t)\nabla_{x_s} \log p(y_{t+1} \mid x_s),$$

where $\lambda$ controls the strength of the guidance. However, solving for the exact $h$-function is often intractable and thus the guided evolution (4) with an approximate control is intrinsically biased (Chidambaram et al., 2024).

In our method, SURGE, we leverage ideas from particle filtering to correct the bias introduced by such approximate guidance. As introduced in Section 2.2, particle filters enforce consistency with the posterior by reweighting and resampling generated particles $\hat{x}_{t+1}^{(m)}$ according to their likelihood $p(y_{t+1} \mid x_{t+1} = \hat{x}_{t+1}^{(m)})$ under the observation $y_{t+1}$. In contrast, our method, SURGE, resamples the full trajectories $x_{t:t+1}$ on the path-measure level, instead of at the endpoint $t + 1$.

### 3.2. Resampling on the Path Space

To this end, we first need to characterize the distribution defined by the guided generation process and quantify its deviation from the desired posterior. By Girsanov's theorem, the Radon-Nikodym derivative (likelihood ratio) between the uncontrolled (or reference) state transition measure $\mathbb{P}$

---

**Algorithm 1** SURGE-Filter: Sequential approximation-free Resampling via Girsanov Estimation

---

**Require:** Initial particles $\{x_0^{(i)}\}_{i=1}^N$, time step $\Delta s$, guidance $G$, variance schedule $\Sigma(t)$, reward $r(\cdot)$, number of steps $T$, resampling threshold $c$.

1: **for** $t = 0$ to $T - 1$ **do**
2:     **for** $k = 0$ to $K - 1$ **do**
3:         **Propagate particle**: $x_{t+s_{k+1}}^{(i)} = x_{t+s_k}^{(i)} + \left(v(x_{t+s_k}^{(i)} \mid x_t) + \Sigma(s_k)\nabla_x G(x_{t+s_k}^{(i)}, s_k \mid y_{t+1})\right)\Delta s + \Sigma^{1/2}(s_k)\sqrt{\Delta s}\,\xi_t^i;$
4:         **Compute weight**: $w_{t+s_{k+1}}^{(i)} = \exp\left((t + s_{k+1})R(x_{t+s_{k+1}}^{(i)}) - (t + s_k)R(X_{t+s_k}^{(i)}) - \Sigma^{1/2}(s_k)\nabla_x G(x_{t+s_k}^{(i)}, s_k \mid\right.$
         $\left. y_{t+1}) \cdot \sqrt{\Delta s}\,\xi_t^i - \frac{1}{2}\Sigma(s_k)\|\nabla_x G(x_{t+s_k}^{(i)}, s_k \mid y_{t+1})\|^2 \Delta s\right)w_{t+s_k}^{(i)};$
5:         **Normalize weights**: $\tilde{w}_{t+s_{k+1}}^{(i)} = w_{t+s_{k+1}}^{(i)} / \sum_{j=1}^N w_{t+s_{k+1}}^{(j)};$
6:         **if** $\widehat{N}_{\text{eff}} = 1/\sum_{i=1}^K (\tilde{w}_{t+s_{k+1}}^{(i)})^2 < c$ **then**
7:             Resample $\{x_{t+s_{k+1}}^{(i)}\}_{i=1}^N \sim \text{Categorical}\left(\{x_{t+s_{k+1}}^{(i)}\}_{i=1}^N, \{\tilde{w}_{t+s_{k+1}}^{(i)}\}_{i=1}^N\right)$ and set $w_{t+s_{k+1}}^{(i)} \leftarrow 1/N$
8:         **end if**
9:     **end for**
10: **end for**

---

and the controlled (or guided) measure $\mathbb{P}^G$ is

$$\frac{d\mathbb{P}(x_{t:t+1}|y_{1:t})}{d\mathbb{P}^G(x_{t:t+1}|y_{1:t+1})} \propto \exp\Bigg($$
$$-\int_0^1 \Sigma^{1/2}(s)\nabla_x G(x_{t+s}, s|y_{t+1}) \cdot dW_s$$
$$-\frac{1}{2}\int_0^1 \left\|\Sigma^{1/2}(s)\nabla_x G(x_{t+s}, s|y_{t+1})\right\|_2^2 ds\Bigg).$$

As introduced earlier, the analysis step aims to sample from the true posterior distribution of states given observations $y_{1:T}$, which is obtained by "tilting" the prior trajectory measure with the observation likelihood:

$$\frac{d\mathbb{P}(x_{t:t+1}|y_{1:t+1})}{d\mathbb{P}(x_{t:t+1}|y_{1:t})} \propto p(y_{t+1} \mid x_{t+1}).$$

For most imperfect control potentials $G$, the path measure $\mathbb{P}^G(\cdot \mid y_{1:t+1})$ generated by controlled simulation does not equal the observation-driven posterior distribution $\mathbb{P}(\cdot \mid y_{1:t+1})$. The discrepancy between them can be characterized using the chain rule for Radon-Nikodym derivatives. For a trajectory $x_{t:t+1}^G$ drawn from the guided simulation distribution $\mathbb{P}^G(\cdot \mid y_{1:t+1})$, the importance weight that corrects it toward the target posterior $\mathbb{P}(\cdot \mid y_{1:t+1})$ is

$$\frac{d\mathbb{P}(x_{t:t+1} \mid y_{1:t+1})}{d\mathbb{P}^G(x_{t:t+1} \mid y_{1:t+1})} = \frac{d\mathbb{P}(x_{t:t+1} \mid y_{1:t+1})}{d\mathbb{P}(x_{t:t+1} \mid y_{1:t})} \frac{d\mathbb{P}(x_{t:t+1} \mid y_{1:t})}{d\mathbb{P}^G(x_{t:t+1} \mid y_{1:t})}$$

$$\propto \exp\Bigg(\log p(y_{t+1} \mid x_{t+1}^G)$$
$$-\int_0^1 \Sigma^{1/2}(s)\nabla_x G(x_{t+s}^G, s \mid y_{t+1}) \cdot dW_s$$
$$-\frac{1}{2}\int_0^1 \left\|\Sigma^{1/2}(s)\nabla_x G(x_{t+s}^G, s \mid y_{t+1})\right\|_2^2 ds\Bigg).$$

$$(5)$$

Similar as the particle filter, we propose to use the weighted empirical measure $\sum_{i=1}^N \tilde{w}_{t+1}^{(i)} \delta_{X_{t+1}^{G,(i)}}$ as approximation-free approximation of the posterior distribution $\mathbb{P}(\cdot \mid y_{1:t+1})$. The construction to weights $w_{t+1}^{(i)}$ is based on the Radon–Nikodym derivative in (5) as

$$w_{t+1}^{(i)} \propto \frac{d\mathbb{P}(x_{t:t+1}^{(i)} \mid y_{1:t+1})}{d\mathbb{P}^G(x_{t:t+1}^{(i)} \mid y_{1:t+1})}.$$

These weights are then normalized to obtain $\tilde{w}_{t+1}^{(i)} = \frac{w_{t+1}^{(i)}}{\sum_{j=1}^N w_{t+1}^{(j)}}$. Resampling is subsequently performed by drawing $N$ particles with replacement from the weighted empirical measure $\sum_{i=1}^N \tilde{w}_{t+1}^{(i)} \delta_{X_{t+1}^{G,(i)}}$, yielding an equally weighted particle approximation of the posterior distribution $p(x_{t+1} \mid y_{1:t+1})$. This resampling step mitigates particle degeneracy and completes a standard SMC update.

### 3.3. Implementation of SURGE

The variance of the resampling step depends on the dispersion of the importance weights. Highly uneven weights lead to large resampling variance and unstable particle behavior. This issue is amplified over long simulations, where accumulated likelihood terms can produce extreme weight scales. To improve stability, we gradually incorporate the likelihood at each time step, thereby controlling weight variability and stabilizing the resampling procedure.

Specifically, during each data assimilation cycle, we discretize the time interval $[0, 1]$ into $K$ uniform subintervals $\{s_k\}_{k=0}^K$, where $s_k = k\Delta s$ and $\Delta s = 1/K$. In each window $[s_k, s_{k+1}]$, we simulate $N$ trajectories $\{x_s^{(i),G}\}_{i=1}^N$ following the guided SDE (4) and apply a resampling step at every assimilation time step for better performance. Small

assimilation time step ensures stable importance weighting and resampling. Within a small time steps, the incremental change in the state $x_s^{(i),G}$ and corresponding reward remains limited.

To make associated Radon-Nikodym derivative varies more smoothly across different particles, we introduce a $s \log p(x_s^{(i),G} | y_{t+1})$ reward to the intermediate generating process. This change leads to the importance weight for window $[s_k, s_{k+1}]$ to

$$
\beta_t^{(i)} = \exp \Bigg( \underbrace{s_{k+1} \log p(y_{s_{k+1}} | x_{s_{k+1}}^{(i),G}) - s_k \log p(x_{s_k} | x_{s_k}^{(i),G})}_{\text{gradually incorporate likelihood}}
$$
$$
- \int_{s_k}^{s_{k+1}} \Sigma(s)^{1/2} \nabla_x G(s_s^{(i),G}, s | y_{t+1}) \cdot \mathrm{d}W_s
$$
$$
- \frac{1}{2} \int_{s_k}^{s_{k+1}} \left\| \Sigma(s)^{1/2} \nabla_x G(s_s^{(i),G}, s | y_{t+1}) \right\|_2^2 \mathrm{d}s \Bigg).
$$

(6)

When $s_k$ and $s_{k+1}$ are sufficiently close, the incremental likelihood term $s_{k+1} \log p \left( y_{s_{k+1}} | x_{s_{k+1}}^{(i),G} \right) - s_k \log p \left( y_{s_k} | x_{s_k}^{(i),G} \right)$ remains small, implying limited variability in the resulting resampling weights and consequently a reduced resampling variance. The full algorithm is shown in Algorithm 1 where we adopt the simple Euler-Maruyama scheme for the computation of the integrals in (6). The approximation-freeness of the SURGE filtering is demonstrated in Appendix A.

# 4. Experiments and Results

In this section, we present several experiments to validate the effectiveness of our proposed method, SURGE. Specifically, we test on the Lorenz system, a forced incompressible Navier-Stokes system, and a real-world large-scale weather forecasting system.

## 4.1. Lorenz System

To validate the effectiveness of SURGE in data assimilation task, we using the Lorenz 1963 system (Lorenz, 1963), a canonical mathematical model originally developed to describe atmospheric convection. Due to its highly nonlinear and chaotic nature, this system is widely established as a rigorous benchmark for evaluating data assimilation and generative modeling techniques (Song et al., 2020; Chen et al., 2025b). The system's state vector, $\mathbf{x} = [x, y, z]^\top$, evolves according to a set of stochastic ordinary differential equations (SDEs) defined as:

$$
\begin{cases}
\dot{x} & = \sigma(y - x) + \xi_x, \\
\dot{y} & = x(\rho - z) - y + \xi_y, \\
\dot{z} & = xy - \beta z + \xi_z,
\end{cases}
$$

*Table 1.* Comparison of data assimilation results on the Lorenz 1963 experiment. SURGE demonstrates consistent performance enhancement across both SDA and FlowDAS backbones, yielding lower RMSE and $W_1$ than all baselines.

| METHOD | RMSE ↓ | $W_1$ ↓ |
|---|---|---|
| BPF (N=20) | 0.0625 | 0.0448 |
| ENKF | 0.0624 | 0.0448 |
| SDA | 0.0589 | 0.0426 |
|   **+ SURGE** | 0.0555 | 0.0396 |
| FLOWDAS | 0.0545 | 0.0388 |
|   **+ SURGE** | **0.0502** | **0.0363** |

We adopt the standard chaotic parameter regime with Prandtl number $\sigma = 10$, Rayleigh number $\rho = 28$, and geometric factor $\beta = 8/3$. The system is subjected to additive Gaussian process noise $\boldsymbol{\xi} = (\xi_x, \xi_y, \xi_z)^\top$, where each component follows $\mathcal{N}(0, 0.05^2)$. To ensure a fair comparison with the baseline FlowDAS (Chen et al., 2025b), we replicate their simulation protocol: the continuous dynamics are integrated using the fourth-order Runge-Kutta (RK4) method, ensuring that any performance variance is attributed to SURGE resampling process rather than numerical discrepancies.

The observation operator $\mathcal{A}$ that acts exclusively on the $x$-coordinate component, while dropping $(y, z)$ coordinate as the sparse observation. The measurement process is governed by: $y_{\text{obs}} = \mathcal{A}(\mathbf{x}) + \eta = \arctan(x) + \eta$, where $\eta$ denotes Gaussian noise with a standard deviation of 0.05.

**Dataset and Experiments.** Following the established experimental design, we generate 1,024 independent trajectories, each comprising 1,024 states. To capture authentic chaotic dynamics, initial states are sampled directly from the system's statistically stationary regime. The dataset is partitioned into training (80%), validation (10%), and evaluation (10%) subsets.

SURGE operates as a training-free, inference-time method, based on baseline setting and model in inference stage. During the testing phase, both methods perform autoregressive generation conditioned on the immediate previous state. We quantitatively evaluate the models by independently estimating 20 trajectories over a prediction horizon of 15 time steps applied in state of art baseline FlowDAS.

**Results.** As demonstrated in Table 1 and Figure 3, our proposed SURGE method consistently outperforms classical filtering baselines (BPF, EnKF), the score-based assimilation method (SDA), and the FlowDAS baseline across all evaluated metrics. The observation conditions in this task are particularly challenging due to an extreme partial bias, where only the noisy $x$-coordinate is observable. Furthermore, these observations are subject to high noise lev-

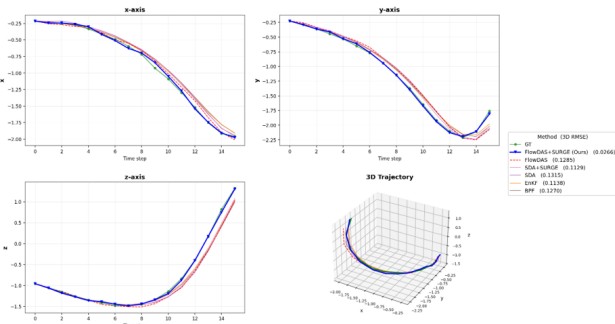

*Figure 3.* Performance comparison between baseline methods and SURGE on the Lorenz system. The SURGE trajectory aligns more closely with the ground-truth trajectory than the baseline trajectories.

els, especially when the ground truth of $x$ approaches zero. Such conditions frequently lead to misleading guidance in gradient-based methods.

### 4.2. Forced Incompressible Navier-Stokes System

To further investigate the capability of SURGE in high-dimensional complex data assimilation tasks, we employed a 2D forced incompressible Navier-Stokes (NS) system driven by random forcing. The physical process is described using the stream function formulation, with the vorticity field $\omega$ serving as the fundamental state variable $x = \omega$. The evolution of the system follows the stochastic partial differential equation:

$$\mathrm{d}\omega + \boldsymbol{v} \cdot \nabla\omega\mathrm{d}t = \nu\Delta\omega\mathrm{d}t - \alpha\omega\mathrm{d}t + \epsilon\mathrm{d}\boldsymbol{\xi}$$

where the velocity field $\boldsymbol{v} = \nabla^{\perp}\psi = (-\partial_y\psi, \partial_x\psi)$ is coupled to the vorticity via the stream function $\psi$, which satisfies the Poisson equation: $-\Delta\psi = \omega$.

The observation process is defined by an obervation operator $\mathcal{A}$ that maps the simulated vorticity fields to a noisy, sparse representation: $\boldsymbol{y} = \mathcal{A}(\omega) + \boldsymbol{\eta}$. In this task, the operator $\mathcal{A}$ linearly downsamples the field or apply sparse observation of partial pixels, while $\boldsymbol{\eta}$ accounts for observational noise.

**Dataset and Experiments.** Following our experimental protocols, we employed JAX-CFD (Kochkov et al., 2021), a GPU-accelerated differentiable simulation framework, to generate high-fidelity datasets. The Navier-Stokes equations were solved via a pseudo-spectral method on a $256^2$ grid with a simulation time step of $\Delta t = 10^{-4}$. Initial conditions were defined by a random vorticity field, $\omega = \nabla \times v$. To ensure physical consistency and eliminate transient effects, the first 50 sampled frames of each trajectory were discarded. We generated 200 trajectories, each providing 200 data frames sampled at intervals of $0.5$ seconds. All snapshots were subsequently downsampled to a resolution of $128^2$. The final dataset was partitioned into training

(80%), validation (10%), and testing (10%) sets. Additionally, a separate out-of-distribution test set consisting of 10 trajectories (1,000 frames each) was generated to evaluate long-term autoregressive performance.

We designed two experimental configurations based on the observation modality: super-resolution and sparse observation. **Super-resolution**: We aimed to reconstruct high-resolution fields $128^2$ from low-resolution inputs of $8^2$. **Sparse Observation**: We reconstructed the complete $128^2$ fluid vorticity field using only 5% of the spatial points as sparse observations. Both experiments utilize $4\times$ temporal downsampling (a 2s interval per step) to evaluate the framework's efficacy with sparse temporal constraints. SURGE utilizes the same pre-trained parameters and architecture as FlowDAS, improve it by optimizing solely at the inference level. For autoregressive prediction, the model is conditioned on a history of $L = 10$ previous states. All models were trained on 80% of the dataset, and evaluations were conducted on 10 unseen trajectories. We evaluate reconstruction using pixel level RMSE for spatial accuracy and kinetic energy spectrum to compute relative error (KES-RE) for physical fidelity. These metrics evaluate model's performance in both pixel-level precision and multi-scale physical characteristics of the flow.

**Results.** As the quantitative results in Figure 4 and Table 2 demonstrate, SURGE maintains a kinetic energy spectrum closer to the ground truth during long-horizon complex flow predictions while simultaneously reducing pixel-level RMSE, outperforming all baselines (BPF, EnKF, SDA, FlowDAS) across both metrics. Qualitative evaluation in Figure 5 further shows that even in the final stages of a 100-step autoregressive rollout, SURGE more accurately captures essential structural features, resulting in significantly higher fidelity compared to the baselines. These results indicate SURGE achieves higher pixel-level precision while maintaining the fundamental multi-scale physical characteristics of the flow.

### 4.3. Weather Forecasting

The accurate weather forecasting is a cornerstone of modern environmental analysis. The highly non-linear and chaotic nature of atmospheric systems make them exceptionally difficulty to simulate by traditional numerical based method. Using learning based surrogate model generally become the essential way in such tasks. While data assimilation provide a promising way to build digital twins for physical weather systems. To evaluate SURGE data assimilation method in this challenging field, we utilized the The Storm EVent ImagRy (SEVIR) (Veillette et al., 2020) and focus on the Vertically Integrated Liquid (VIL) product as a key value for weather forecasting.

**Dataset and Experiments.** To provide a rigorous valida-

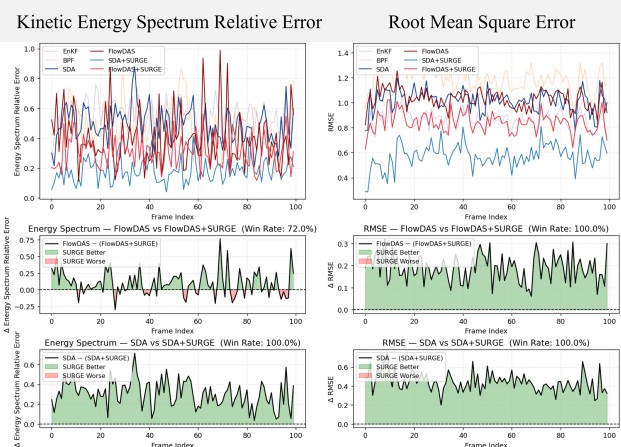

*Figure 4.* Performance comparison between baseline methods and SURGE on a forced Navier-Stokes system. Relative kinetic energy spectrum error (left) and RMSE (right) over 100 inference frames. The top row compares all baselines (EnKF, BPF, SDA, FlowDAS) against SDA+SURGE and FlowDAS+SURGE, while the middle and bottom rows show the per-frame improvement of SURGE over FlowDAS and SDA backbones respectively, with win rates reported in each subtitle. SURGE demonstrates consistent performance enhancement across both backbones in this high-dimensional chaotic flow task.

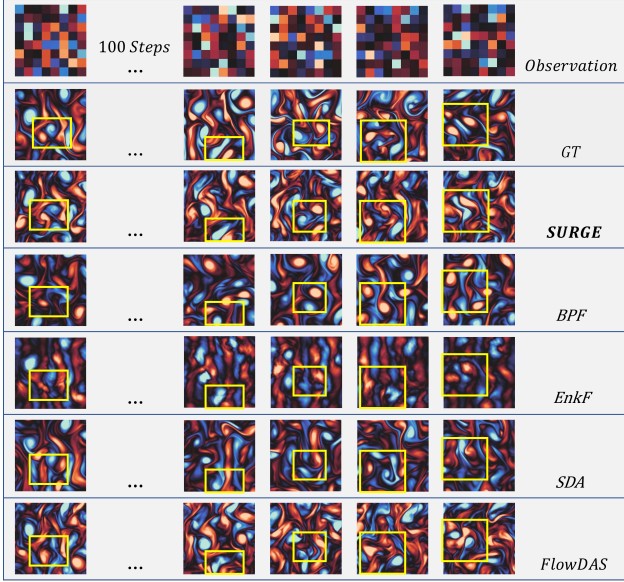

*Figure 5.* Qualitative comparison of vorticity field reconstruction between baseline methods and SURGE. Positive red values indicate clockwise rotation, and negative blue values indicate counter-clockwise rotation. Yellow boxes highlight regions where SURGE better recovers physical structures under large temporal gaps and sparse observations.

tion of our method, we adopt the experimental framework established by FlowDAS (Chen et al., 2025b) a recent state-of-the-art approach in data assimilation. We utilize the SEVIR dataset, focusing on the Vertically Integrated Liquid (VIL). Following the FlowDAS setting, we emulate the harsh conditions of real-world sensing by randomly sampling only 10% of the grid cells. Model use information from $L = 6$ historical frames to predict the state at $t + 10$ min, effectively testing its capacity to handle both extreme data sparsity and complex physical evolution. We utilize SURGE method in FlowDAS inference step. Following the aviation weather standards established by (Robinson et al., 2002), we evaluate our model across 2 thresholds of $\tau_{20}, \tau_{40}$ dBZ, capturing both general coverage and high-intensity extremes. The Critical Success Index (CSI) and RMSE are used as metrics to account for both hits and false alarms in the predictions.

**Results.** As shown in Table 3, SURGE enhances the Flow-DAS inference process, achieving lower RMSE and higher CSI scores at target thresholds than all baselines (BPF, EnKF, SDA, FlowDAS). Qualitative comparisons in Figure 6 further demonstrate that SURGE recovers more coherent structures with sharper boundaries, as well as accurate localization and intensity. These results confirm that SURGE effectively improves data assimilation across both SDA and FlowDAS backbones without additional training or data, highlighting its capability in complex real-world physical evolution tasks without explicit equations.

## 5. Conclusion and Future Work

We introduce SURGE, a training-free, plug-and-play data assimilation framework that enhances any diffusion surrogate under sparse, noisy measurements. By leveraging particle filtering during inference, SURGE approximation-freely integrates observations to rectify predictions in high-dimensional, nonlinear dynamical systems. Experiments on Lorenz systems, Navier-Stokes flow, and weather forecasting demonstrate significant performance gains, particularly in challenging tasks like extreme super-resolution and long-term autoregressive prediction under sparse data and rapid dynamic shifts. SURGE serves as a universal online enhancement, effectively correcting diffusion-based sequence models without retraining. However, its efficacy is intrinsically linked to the base model's quality and remains sensitive to hyperparameters. Future work will focus on developing robust reward and guidance mechanisms to accommodate multi-scale inputs and diverse guidance types, further improving system stability in complex physical simulations.

## Impact Statement

This paper presents work whose goal is to advance the field of machine learning. There are many potential societal consequences of our work, none of which we feel must be specifically highlighted here.

*Table 2.* Comparison of Navier-Stokes results on **super-resolution** ($8^2 \rightarrow 128^2$) and **sparse recovery** ($5\% \rightarrow 100\%$). SURGE demonstrates consistent performance enhancement across both SDA and FlowDAS backbones, outperforming all baselines.

| METHOD | KES-RE↓ | RMSE↓ |
|---|---|---|
| *Super-resolution* ($8^2 \rightarrow 128^2$) | | |
| BPF (N=20) | 0.490 | 1.143 |
| ENKF | 0.551 | 0.847 |
| SDA | 0.473 | 0.987 |
| + **SURGE** | 0.417 | 0.966 |
| FLOWDAS | 0.401 | 1.018 |
| + **SURGE** | **0.317** | **0.851** |
| *Sparse recovery* ($5\% \rightarrow 100\%$) | | |
| BPF (N=20) | 0.486 | 1.133 |
| ENKF | 0.676 | 0.800 |
| SDA | 0.231 | 0.590 |
| + **SURGE** | **0.207** | **0.564** |
| FLOWDAS | 0.543 | 0.872 |
| + **SURGE** | 0.278 | 0.673 |

*Table 3.* Comparison of data assimilation results on the Weather forecast dataset. SURGE demonstrates consistent performance enhancement across both SDA and FlowDAS backbones, yielding lower RMSE and higher CSI scores than all baselines.

| METHOD | RMSE ↓ | CSI($\tau_{20}$)↑ | CSI($\tau_{40}$)↑ |
|---|---|---|---|
| BPF (N=20) | 0.0939 | 0.4146 | 0.2171 |
| ENKF | 0.1281 | 0.4970 | 0.3150 |
| SDA | 0.3511 | 0.2757 | 0.1569 |
| + **SURGE** | 0.0925 | 0.4089 | 0.2196 |
| FLOWDAS | 0.0657 | 0.5779 | 0.4044 |
| + **SURGE** | **0.0513** | **0.6197** | **0.4541** |

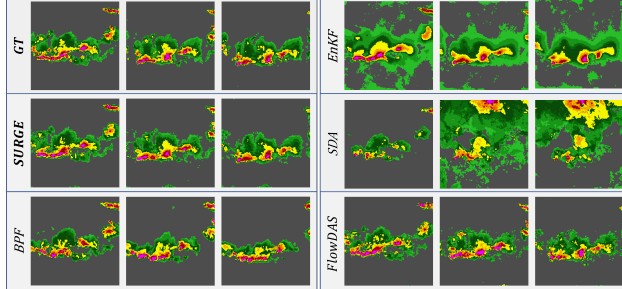

*Figure 6.* Qualitative and quantitative comparison of precipitation forecasting between baseline methods and SURGE. FlowDAS tends to produce blurry predictions, failing to maintain high-intensity echo regions. In contrast, SURGE effectively preserves sharp boundaries and fine-grained patterns, achieving consistently higher CSI scores across all thresholds.

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

# A. Theoretical Analysis: approximation-freeness of the `SURGE` Filter

This appendix formalizes the correctness claim behind SURGE: although the guided sampler (4) generally does *not* sample from the filtering posterior, it can be used as a *proposal* on path space, and the Girsanov-corrected importance weights in (5) exactly map the proposal back to the desired posterior. In particular, SURGE introduces no systematic bias from using an approximate guidance potential $G$; $G$ only affects proposal efficiency and weight variance.

## A.1. Posterior as a tilted path measure

Fix an assimilation step $t \to t+1$. Recall the uncontrolled diffusion surrogate transition (3), whose induced path measure on $C([t, t+1], \mathbb{R}^D)$ we denote by $\mathbb{P}(\mathrm{d}x_{t:t+1} \mid y_{1:t})$. The guided sampler (4) induces a second path measure, denoted $\mathbb{P}^G(\mathrm{d}x_{t:t+1} \mid y_{1:t+1})$.

Let the (physical) observation likelihood be

$$\ell_{t+1}(x) := p(y_{t+1} \mid x).$$

The filtering posterior path measure is obtained by tilting the prior path measure by the terminal likelihood:

$$\mathbb{P}(\mathrm{d}x_{t:t+1} \mid y_{1:t+1}) = \frac{1}{Z_{t+1}} \ell_{t+1}(x_{t+1}) \mathbb{P}(\mathrm{d}x_{t:t+1} \mid y_{1:t}), \qquad Z_{t+1} := \mathbb{E}_{\mathbb{P}(\cdot \mid y_{1:t})}[\ell_{t+1}(X_{t+1})]. \tag{7}$$

The filtering distribution $p(x_{t+1} \mid y_{1:t+1})$ is the marginal of (7) at time $t+1$.

## A.2. Girsanov change-of-measure and the `SURGE` weight

Define the (whitened) control associated with the guidance potential $G$ by

$$u_s(x) := \Sigma^{1/2}(s) \nabla_x G(x, s \mid y_{t+1}), \qquad s \in [0, 1].$$

Assume the standard Novikov condition (sufficient for Girsanov) holds under the guided measure:

$$\mathbb{E}_{\mathbb{P}^G(\cdot \mid y_{1:t+1})} \left[ \exp\left( \frac{1}{2} \int_0^1 \|u_s(X_{t+s}^G)\|_2^2 \, \mathrm{d}s \right) \right] < \infty. \tag{8}$$

Then $\mathbb{P}(\cdot \mid y_{1:t})$ is absolutely continuous with respect to $\mathbb{P}^G(\cdot \mid y_{1:t+1})$, and Girsanov's theorem yields the Radon–Nikodym derivative

$$\frac{\mathrm{d}\mathbb{P}(\cdot \mid y_{1:t})}{\mathrm{d}\mathbb{P}^G(\cdot \mid y_{1:t+1})}(x_{t:t+1}) = \exp\left( -\int_0^1 u_s(x_{t+s}) \cdot \mathrm{d}W_s - \frac{1}{2} \int_0^1 \|u_s(x_{t+s})\|_2^2 \, \mathrm{d}s \right), \tag{9}$$

where $W$ is the Brownian motion under the guided measure $\mathbb{P}^G(\cdot \mid y_{1:t+1})$.

Combining (7) and (9), the *unnormalized* importance weight for a guided trajectory $X_{t:t+1}^G \sim \mathbb{P}^G(\cdot \mid y_{1:t+1})$ is

$$w_{t+1}(X_{t:t+1}^G) := \ell_{t+1}(X_{t+1}^G) \frac{\mathrm{d}\mathbb{P}(\cdot \mid y_{1:t})}{\mathrm{d}\mathbb{P}^G(\cdot \mid y_{1:t+1})}(X_{t:t+1}^G), \tag{10}$$

which is exactly the continuous-time counterpart of (5) (up to the normalizing constant $Z_{t+1}$).

**Proposition A.1** (approximation-freeness for the unnormalized posterior functional). *Let* $\Phi : C([t, t+1], \mathbb{R}^D) \to \mathbb{R}$ *be bounded and measurable. Under* (8),

$$\mathbb{E}_{\mathbb{P}^G(\cdot \mid y_{1:t+1})}\left[ w_{t+1}(X_{t:t+1}^G) \, \Phi(X_{t:t+1}^G) \right] = \mathbb{E}_{\mathbb{P}(\cdot \mid y_{1:t})}\left[ \ell_{t+1}(X_{t+1}) \, \Phi(X_{t:t+1}) \right] = Z_{t+1} \, \mathbb{E}_{\mathbb{P}(\cdot \mid y_{1:t+1})}\left[ \Phi(X_{t:t+1}) \right]. \tag{11}$$

*In particular, taking* $\Phi(x_{t:t+1}) = \phi(x_{t+1})$ *gives*

$$\mathbb{E}_{\mathbb{P}^G}\left[ w_{t+1} \, \phi(X_{t+1}^G) \right] = Z_{t+1} \, \mathbb{E}\left[ \phi(X_{t+1}) \mid y_{1:t+1} \right].$$

*Proof.* The first equality is the defining property of the Radon–Nikodym derivative in (9), and the second equality follows immediately from the tilted-measure definition (7). $\square$

**Corollary A.2** (Asymptotic exactness of self-normalized SURGE estimates). *Let $X_{t:t+1}^{G,(1)}, \ldots, X_{t:t+1}^{G,(N)} \overset{i.i.d.}{\sim} \mathbb{P}^G(\cdot \mid y_{1:t+1})$ and define the self-normalized estimator*

$$\hat{\pi}_{t+1}^N(\phi) := \sum_{i=1}^N \tilde{w}_{t+1}^{(i)} \phi\left(X_{t+1}^{G,(i)}\right), \qquad \tilde{w}_{t+1}^{(i)} := \frac{w_{t+1}^{(i)}}{\sum_{j=1}^N w_{t+1}^{(j)}}.$$

*Then $\hat{\pi}_{t+1}^N(\phi) \to \mathbb{E}[\phi(X_{t+1}) \mid y_{1:t+1}]$ in probability as $N \to \infty$.*

*Proof.* This is the standard self-normalized importance sampling law of large numbers, applied with the target measure (7) and weight (10); see Proposition A.1. $\square$

### A.3. Gradual likelihood incorporation as an exact telescoping factorization

Algorithm 1 updates weights along the internal diffusion time $s \in [0, 1]$ on a grid $0 = s_0 < s_1 < \cdots < s_K = 1$, with $\Delta s = s_{k+1} - s_k$.

Define the log-likelihood reward

$$R(x) := \log \ell_{t+1}(x) = \log p(y_{t+1} \mid x).$$

The gradual update used in SURGE is an *exact factorization* of the terminal likelihood $\ell_{t+1}(x_{t+1})$:

**Lemma A.3** (Telescoping likelihood decomposition). *For any trajectory $x_{t:t+1}$ and any grid $0 = s_0 < \cdots < s_K = 1$,*

$$\ell_{t+1}(x_{t+1}) = \prod_{k=0}^{K-1} \exp\left(s_{k+1} R(x_{t+s_{k+1}}) - s_k R(x_{t+s_k})\right). \tag{12}$$

*Proof.* Summing the exponents gives

$$\sum_{k=0}^{K-1} \left(s_{k+1} R(x_{t+s_{k+1}}) - s_k R(x_{t+s_k})\right) = s_K R(x_{t+s_K}) - s_0 R(x_{t+s_0}) = R(x_{t+1}),$$

since all intermediate terms cancel and $s_0 = 0$, $s_K = 1$. $\square$

Thus, distributing the likelihood contribution across internal steps does *not* change the final target: the product of incremental likelihood factors remains exactly $\ell_{t+1}(X_{t+1})$.

### A.4. Discrete-time incremental weights (Euler–Maruyama)

Let $\Delta W_k := W_{s_{k+1}} - W_{s_k}$ so that $\Delta W_k = \sqrt{\Delta s}\, \xi_k$ with $\xi_k \sim \mathcal{N}(0, I)$. Approximating the integrals in (9) by Euler–Maruyama, and using Lemma A.3, yields the per-step incremental weight

$$\beta_{t,k}^{(i)} = \exp\left(s_{k+1} R(X_{t+s_{k+1}}^{G,(i)}) - s_k R(X_{t+s_k}^{G,(i)}) - u_{s_k}(X_{t+s_k}^{G,(i)}) \cdot \sqrt{\Delta s}\, \xi_k^{(i)} - \tfrac{1}{2} \|u_{s_k}(X_{t+s_k}^{G,(i)})\|_2^2\, \Delta s\right), \tag{13}$$

and the recursion $w_{t+s_{k+1}}^{(i)} = w_{t+s_k}^{(i)} \beta_{t,k}^{(i)}$, where $u_{s_k} = \Sigma^{1/2}(s_k) \nabla_x G(\cdot, s_k \mid y_{t+1})$. This is the discrete analogue of (10) and matches Algorithm 1.

### A.5. Resampling preserves expectations

Resampling is used to control weight degeneracy; it does not change the represented distribution in expectation.

**Lemma A.4** (Conditional approximation-freeness of multinomial resampling). *Let $\{(X^{(i)}, \tilde{w}^{(i)})\}_{i=1}^N$ be weighted particles with $\sum_i \tilde{w}^{(i)} = 1$. Let $\{\tilde{X}^{(i)}\}_{i=1}^N$ be obtained by multinomial resampling from $\sum_i \tilde{w}^{(i)} \delta_{X^{(i)}}$ and assigning equal weights $1/N$. Then for any integrable test function $\phi$,*

$$\mathbb{E}\left[\frac{1}{N} \sum_{i=1}^N \phi(\tilde{X}^{(i)}) \,\middle|\, \{(X^{(j)}, \tilde{w}^{(j)})\}_{j=1}^N\right] = \sum_{j=1}^N \tilde{w}^{(j)}\, \phi(X^{(j)}).$$

*Proof.* Conditioned on the current weighted particles, the resampling indices are i.i.d. with $\mathbb{P}(A^{(i)} = j) = \tilde{w}^{(j)}$ and $\tilde{X}^{(i)} = X^{(A^{(i)})}$, hence $\mathbb{E}[\phi(\tilde{X}^{(i)}) \mid \cdot] = \sum_j \tilde{w}^{(j)} \phi(X^{(j)})$. Averaging over $i$ gives the claim. □

*Remark* A.5. Lemma A.4 also holds for standard low-variance schemes (residual, stratified, systematic), which are approximation-free in the same conditional sense.

### A.6. Takeaway

Proposition A.1 and Corollary A.2 show that SURGE is an importance sampler on diffusion-time path space whose target is exactly the filtering posterior (7). Lemma A.3 shows that gradual likelihood incorporation is an exact factorization of the same terminal likelihood (hence it does not alter the target), and Lemma A.4 shows resampling preserves expectations conditional on the current weighted particles. Therefore, approximate guidance affects only efficiency and variance, not the target posterior in the large-particle limit.

## B. Additional Experimental Details

### B.1. Datasets and Problem Settings

**Lorenz System.** We evaluate the framework on the Lorenz 1963 system, a chaotic low-dimensional dynamical system widely used in data assimilation benchmarks. The state vector $x = (a, b, c)^\top \in \mathbb{R}^3$ evolves according to the following stochastic ordinary differential equations (ODEs):

$$\begin{cases} \dfrac{\mathrm{d}a}{\mathrm{d}t} &= \mu(b - a) + \xi_1, \\ \dfrac{\mathrm{d}b}{\mathrm{d}t} &= a(\rho - c) - b + \xi_2, \\ \dfrac{\mathrm{d}c}{\mathrm{d}t} &= ab - \tau c + \xi_3, \end{cases}$$

where the system parameters are set to standard values: $\mu = 10$, $\rho = 28$, and $\tau = 8/3$. The process noise $\xi = (\xi_1, \xi_2, \xi_3)^\top$ consists of Gaussian components, each with a standard deviation of $\sigma = 0.25$. Due to the chaotic nature of the system, we employ the fourth-order Runge-Kutta (RK4) method for numerical simulation. The deterministic state updates from time $t_n$ to $t_{n+1} = t_n + h$ are computed using standard RK4 intermediate steps ($k_1$ through $k_4$) , followed by the addition of the stochastic force. Dataset: We generated 1,024 independent trajectories, each consisting of 1,024 time steps. The data was split into training (80%), validation (10%), and evaluation (10%) sets. Initial states were sampled from the system's statistically stationary regime. The observation operator $\mathcal{A}(\cdot)$ transforms the first state component $a$ using an arctangent function. The observation $y_k$ at time step $k$ is given by:

$$y_k = \mathcal{A}(x_k) + \eta_k = \arctan(a_k) + \eta_k,$$

where $\eta_k$ represents observation noise. In our experimental setting, $\eta_k$ is drawn from a zero-mean Gaussian distribution $\mathcal{N}(0, \gamma^2)$ with a standard deviation of $\gamma = 0.05$.

**Model Architecture and Training.** To ensure a fair comparison, we adopted identical experimental settings for both methods. Specifically, all drift model configurations and guidance parameters in SURGE are kept consistent with those used in FlowDAS.

For this low-dimensional task, the drift velocity field $b_s(X_s, X_0)$ is approximated using a fully connected neural network (MLP). The architecture details are as follows: Hidden Layers: 5 layers with a hidden dimension of 256 each. Input/Output: Input and output dimensions are both 3 (corresponding to the state space). Conditioning: The initial state condition $X_0$ and the interpolation time $s$ are injected via embeddings of dimension 4. Training: The model is optimized using the Adam optimizer with a base learning rate of 0.005. A linear learning rate scheduler is applied, and the training is conducted for 5000 epochs. During the inference stage, we utilize the following hyperparameters: Monte Carlo Sampling ($J$): 21 Sampling Step Size ($\zeta$): 0.0002. This step size is set smaller than in other experiments to accurately capture the sensitive dynamics of the chaotic Lorenz system

**Navier-stokes Flow System.** We assess the proposed method on the high-dimensional 2D incompressible Navier-Stokes (NS) equations defined on a torus $\mathbb{T}^2 = [0, 2\pi]^2$. The system dynamics for the vorticity field $\omega(x, t)$ are governed by the

stochastic partial differential equation :

$$dw + (v \cdot \nabla \omega)dt = \nu \Delta \omega dt - \alpha \omega dt + \epsilon d\xi, \tag{14}$$

where $v$ is the velocity field related to vorticity by $\omega = \nabla \times v$. The system parameters are set to viscosity $\nu = 10^{-3}$, damping coefficient $\alpha = 0.1$, and noise magnitude $\epsilon = 1$. The stochastic forcing term $d\xi$ acts on specific Fourier modes.

**Data Generation and Observation.** The ground truth trajectories are generated using a pseudo-spectral method with a spatial resolution of $256^2$ and a time step of $\Delta t = 10^{-4}$. For training and evaluation, the simulation snapshots are downsampled to a resolution of $128^2$ and recorded at intervals of $\Delta t = 0.5$. The observation model assumes the measurements $y$ are corrupted by Gaussian noise:

$$y = \mathcal{A}(\omega) + \eta, \quad \eta \sim \mathcal{N}(0, \gamma^2 I), \tag{15}$$

where the noise standard deviation is set to $\gamma = 0.05$.

**Model Architecture and Training.** To ensure a fair comparison, we adopted identical experimental settings for both methods. Specifically, all drift model configurations and guidance parameters in SURGE are kept consistent with those used in FlowDAS. We adopt a U-Net architecture to approximate the drift velocity $b_s(X_s, X_0)$, following the configuration in FlowDAS. The network conditions on the past $L = 10$ states via channel concatenation. The architecture features an initial channel dimension of 128 with channel multipliers of $(1, 2, 2, 2)$ at subsequent stages . It incorporates group normalization (8 groups) and self-attention mechanisms (4 heads, 64 dimensions per head) to capture multi-scale features . The model is trained using the AdamW optimizer with a cosine annealing learning rate schedule, starting from a base learning rate of $2 \times 10^{-4}$ . The training process runs for 1,000 epochs with a batch size of 32.

**Weather Forecasting.** We evaluate the method on the Storm EVent Imagery and Radar (SEVIR) dataset, a large-scale meteorological dataset containing diverse storm events. We focus specifically on the Vertically Integrated Liquid (VIL) product, which serves as a 2D proxy for precipitation intensity. Each data sample consists of a $128 \times 128$ grid representing a $384\text{km} \times 384\text{km}$ area, with a temporal resolution of 10 minutes. The forecasting task involves conditioning on $L = 6$ past frames (from $t - 50$ min to $t$ min) to predict the future state at $t + 10$ min. The observations are sparse and noisy. We emulate sparse radar coverage by randomly sampling 10% of the grid cells as available measurements. We assume the observations are corrupted by Gaussian noise with a standard deviation of $\gamma = 0.05$.

**Model Architecture and Training.** Although the original study suggests using a Fourier Neural Operator for this task, the official FlowDAS implementation currently only provides the U-Net architecture. To ensure alignment with the available codebase, we employ the U-Net as the backbone for the drift velocity model $b_s(X_s, X_0)$. The network is configured with an initial channel dimension of 128 and channel multipliers of $(1, 2, 2, 2)$ across stages. It incorporates ResNet blocks with group normalization (8 groups) and self-attention mechanisms (4 heads, 64 dimensions per head). The network inputs include the concatenation of the previous $L = 6$ states and the current interpolation state $X_s$, with the time step $s$ injected via learned sinusoidal embeddings. We observed that retraining the model from scratch using the provided open-source code did not yield acceptable convergence or performance levels. Therefore, to ensure a valid assessment of the framework's capabilities, we directly employed the official pre-trained model weights provided by the FlowDAS authors for this task. During the inference stage, we perform the stochastic generation with $J = 25$ Monte Carlo sampling steps and a sampling step size of $\zeta = 0.1$.

### B.2. Implementation Details

**Diffusion Model Architecture.** We employ a U-Net style architecture for the drift model, a design fully adopted from the FlowDAS framework. It is important to note that the input and output configurations vary across the different tasks (i.e., Lorenz, Navier-Stokes, and Weather Forecasting). Specific details regarding these task-dependent settings are provided in the *Datasets and Problem Settings* section.

**Guidance Term.** For the guidance term $G$, we employ the gradient-guidance mechanism introduced in FlowDAS. This method augments the learned drift term $b_s(X_s, X_0)$ with an observation-dependent correction term during the inference process:

$$b_s(X_s, y, X_0) = b_s(X_s, X_0) + \frac{\nabla \log p(y|X_s, X_0)}{\lambda_s \beta_s}. \tag{16}$$

As the likelihood term $\nabla \log p(y|X_s, X_0)$ is analytically intractable, it is approximated via Monte Carlo marginalization

using $J$ posterior samples:

$$\nabla \log p(y|X_s, X_0) \approx \sum_{j=1}^{J} w_j \nabla \log p(y|\hat{X}_1^{(j)}), \tag{17}$$

where $w_j$ denotes the softmax-normalized likelihood weights derived from the observation error $\|y - \mathcal{A}(\hat{X}_1^{(j)})\|_2^2$. To ensure a strictly fair comparison and isolate the contributions of our proposed improvements, we adopt this guidance strategy and its associated hyperparameters identically to the FlowDAS setting.

**Reward Term.** We employ a reward mechanism to modulate particle weights, steering the generation towards high-probability states. The specific form of the reward function $r(x)$ depends on the task. For standard observation consistency (likelihood mode), we define the reward based on the measurement residual:

$$r(x) = -\frac{1}{2} \sum \left(\lambda(y - \mathcal{A}(x))\right)^2, \tag{18}$$

where $y$ is the observation and $\mathcal{A}(\cdot)$ is the observation operator.

**Hyperparameters.**

- Number of particles: $N = 3$ for Lorenz system, $N = 4$ for Navier-stokes flow and $N = 4$ for Weather forecasting.

- Resampling threshold (effective sample size): $c = 0.75N$ for Lorenz system, $c = 0.5N$ for Navier-stokes flow and $c = 0.5N$ for weather forecasting.

- Diffusion coefficient: $\Sigma(s) = \sigma^2 I$ with $\sigma = 0.1$

- Learning rate: $2 \times 10^{-5}$ for Lorenz, $2 \times 10^{-5}$ for Navier-stokes flow and $2 \times 10^{-5}$ for Weather forecasting.

- Euler-Maruyama (Diffusion) steps: $600$ for Lorenz, $20$ for Navier-stokes and $500$ for weather forecasting.

### B.3. Evaluation Metrics

- **Root Mean Square Error (RMSE).**

$$\text{RMSE} = \sqrt{\frac{1}{TD} \sum_{t=1}^{T} \|\hat{x}_t - x_t^{\text{true}}\|^2},$$

where $\hat{x}_t$ is the posterior mean estimate.

- **Relative Error of Kinetic Energy Spectrum (KES-RE).**

$$\text{KES-RE} = \frac{\|\hat{E}(k) - E^{\text{true}}(k)\|_2}{\|E^{\text{true}}(k)\|_2},$$

Here, $E(k)$ represents the kinetic energy spectrum, calculated by summing the energy of Fourier modes within concentric shells of wavenumber $k$:

$$E(k) = \sum_{k \leq \|\mathbf{k}\| < k+1} \frac{1}{2} \|\hat{\mathbf{v}}(\mathbf{k})\|^2,$$

- **Critical Success Index (CSI).**

$$\text{CSI}(\tau) = \frac{\text{Hits}}{\text{Hits} + \text{Misses} + \text{False Alarms}},$$

where hits, misses, and false alarms are calculated based on a binary map generated by an intensity threshold $\tau$. The threshold $\tau$ signifies the severity of the weather event: a lower $\tau$ (e.g., $\tau_{20}$) assesses the prediction of general rainfall, while a higher $\tau$ (e.g., $\tau_{40}$) specifically evaluates the model's ability to capture extreme events, such as heavy storms.

*Table 4.* Full results on the Lorenz 1963 experiment. SURGE consistently improves both SDA and FlowDAS backbones.

| METHOD | RMSE $\downarrow$ | $W_1 \downarrow$ |
|---|---|---|
| BPF (N=20) | 0.0625 | 0.0448 |
| DM | 0.0766 | 0.0549 |
| ENKF | 0.0624 | 0.0448 |
| SDA | 0.0589 | 0.0426 |
| + **SURGE** | 0.0555 | 0.0396 |
| FLOWDAS | 0.0545 | 0.0388 |
| FLOWDAS AVG | 0.0923 | 0.0698 |
| + **SURGE** | **0.0502** | **0.0363** |

*Table 5.* Full results on Navier-Stokes super-resolution (SR, $8^2 \to 128^2$) and sparse observation (SO, $5\% \to 100\%$). textttSURGE consistently improves both SDA and FlowDAS backbones.

| | SR ($8^2 \to 128^2$) | | SO ($5\% \to 100\%$) | |
|---|---|---|---|---|
| METHOD | KES-RE $\downarrow$ | RMSE $\downarrow$ | KES-RE $\downarrow$ | RMSE $\downarrow$ |
| BPF (N=20) | 0.490 | 1.143 | 0.486 | 1.133 |
| DM | 0.657 | 1.310 | 0.663 | 1.320 |
| ENKF | 0.551 | 0.847 | 0.676 | 0.800 |
| SDA | 0.473 | 0.987 | 0.231 | 0.590 |
| + **SURGE** | 0.417 | 0.966 | **0.207** | **0.564** |
| FLOWDAS | 0.401 | 1.018 | 0.543 | 0.872 |
| FLOWDAS AVG | 0.329 | 0.898 | 0.315 | 0.723 |
| + **SURGE** | **0.317** | **0.851** | 0.278 | 0.673 |

## B.4. Additional Results

**Baselines.** We compare SURGE against several representative data assimilation methods spanning classical filtering and modern generative approaches. For a fair comparison, BPF, EnKF, and SDA all share the same learned dynamics model as the forecast drift.

**Bootstrap Particle Filter (BPF)** is a classical sequential Monte Carlo method that propagates weighted particles through the dynamics and reweights them by the observation likelihood(Gordon et al., 1993); we use $N = 20$.

**Diffusion Model (DM)** refers to a plain diffusion sampler that generates trajectories from the learned prior without observation guidance, included to isolate the contribution of guidance.

**Ensemble Kalman Filter (EnKF)** maintains a finite ensemble and applies a Kalman-style update under a Gaussian approximation of the forecast distribution(Evensen, 2003).

**Score-based Data Assimilation (SDA)** learns a score model of the trajectory prior and conditions on observations via score-based posterior sampling(Rozet & Louppe, 2023).

**FlowDAS** is the current state-of-the-art flow-matching framework for data assimilation, which builds a guided proposal from observations(Chen et al., 2025b); **FlowDAS AVG** denotes the average of multiple independent FlowDAS runs. As the strongest existing baseline, FlowDAS is also the primary backbone we plug SURGE into.

**Lorenz.** Additional visualization results of the state trajectories for the Lorenz experiment are presented in Figure 7 and Figure 8. It can be clearly figure out that by optimizing inference step with SURGE, FlowDAS baseline achieves better results. Full quantitative results are reported in Table 4.

**Navier-stokes.** Additional results of the trajectories wise performacne for the Navier-stokes experiment are presented in Figure 11 and Figure 12. SURGE Outperform in more steps accurate kinetic energy spectrum with ground truth and smaller root mean square error in these results. Full quantitative results are reported in Table 5.

*Table 6.* Full results on the Weather forecasting task. SURGE consistently improves both SDA and FlowDAS backbones.

| METHOD | RMSE $\downarrow$ | CSI($\tau_{20}$) $\uparrow$ | CSI($\tau_{40}$) $\uparrow$ |
|---|---|---|---|
| BPF (N=20) | 0.0939 | 0.4146 | 0.2171 |
| DM | 0.1259 | 0.3181 | 0.1486 |
| ENKF | 0.1281 | 0.4970 | 0.3150 |
| SDA | 0.3511 | 0.2757 | 0.1569 |
| + **SURGE** | 0.0925 | 0.4089 | 0.2196 |
| FLOWDAS | 0.0657 | 0.5779 | 0.4044 |
| FLOWDAS AVG | 0.0534 | 0.6163 | 0.4477 |
| + **SURGE** | **0.0513** | **0.6197** | **0.4541** |

**Weather forecasting.** Additional qualitative and quantitative results comparison for the weather forecasting experiment are presented in Figure 14, Figure 15, Figure 16 and Figure 17. SURGE outperforms the BASE model by generating more accurate VIL intensity patterns relative to the ground truth and achieving consistently higher Critical Success Index (CSI) scores across various thresholds and time steps in these results. Full quantitative results are reported in Table 6. Regarding the specific pixel intensity thresholds (Th) employed in the CSI analysis over the $[0, 255]$ range: The threshold parameter $\tau = 0.3$ used in our method corresponds approximately to a normalized value of 0.2 in the $[0, 1]$ interval, which maps to Th=74 (representing moderate intensity). The broadest evaluation threshold, Th=16, captures the lower bound of detectable precipitation. Higher thresholds evaluate performance on increasingly severe weather: Th=133 indicates high intensity, Th=160 and Th=181 represent very high intensities, and Th=219 corresponds to extreme intensity events, matching the VIL Intensity color scale shown in the figures.

### B.5. Analysis

**Dependency in Diffusion surrogate.** While SURGE demonstrates superior performance in most regimes, we observe a specific failure mode in the Lorenz system under partial observation (observing only the $x$-coordinate). As illustrated in Figure 9, when the trajectory initializes near the null-isocline ($x \approx 0$), the learned drift model (diffusion surrogate) exhibits instability, leading to oscillatory behaviors (Figure 9, Left). In this scenario, applying SURGE can result in high-variance estimates. Although SURGE correctly attempts to steer the particles towards the observation (green dashed line), the aggressive re-weighting mechanism discards alternative paths. Given the poor quality of the proposal distribution in this specific region, this selection bias leads to trajectory overshooting and deviation from the true state (Figure 9, Right). This can be further explained by Figure 10.

This case highlights a fundamental limitation: SURGE performance is intrinsically coupled with the stability of the underlying drift model. Unlike image generation tasks where semantic guidance allows for diverse outputs that remain perceptually valid, physical forecasting requires precise state recovery.

In simple regimes where the diffusion surrogate is highly confident (i.e., generated samples are nearly identical), SURGE becomes redundant as the proposal distribution already collapses to a deterministic path. In complex regimes where the surrogate exhibits necessary stochasticity, SURGE acts as a de-biasing mechanism. However, under limited particle counts, if the proposal distribution (drift) is fundamentally misaligned or unstable (as seen in the $x \approx 0$ case), the re-sampling process may exacerbate errors rather than correct them. Carefully balancing the output stochasticity (temperature) of the diffusion surrogate with the strength of the SURGE guidance is a non-trivial optimization problem. This is particularly challenging in physical tasks where evaluation is based on deterministic accuracy rather than perceptual diversity. Future work will focus on developing adaptive mechanisms to robustly handle such edge cases without manual parameter tuning.

**Ensemble Smoothing Dilemma.** SURGE inherently generates a particle ensemble to approximate the posterior. While this strategy proves highly effective for the Lorenz and Weather forecasting tasks, it introduces a smoothing effect in the Navier-Stokes experiment, where averaging the particle cloud dampens high-frequency flow details. Nevertheless, preserving reasonable details within an ensemble output remains a challenge, as the particle filter is fundamentally designed to output a distribution. Despite the visual smoothing, quantitative analysis demonstrates that the SURGE ensemble output significantly surpasses both the FlowDAS baseline and individual particle trajectories in error metrics. This superiority validates the core effectiveness of our particle filter framework. It is important to note that this trade-off between spectral detail and RMSE is unique to the high-frequency nature of Navier-Stokes flow and is not observed in the other experimental

*Table 7.* Effective sample size (ESS/$K$) of SURGE across tasks and backbones, evaluated at $N = 4$ and $N = 100$ particles. ESS decreases as $N$ grows, consistent with weight collapse, but SURGE maintains substantially non-degenerate weights even at $N = 100$.

| | $N = 4$ | | $N = 100$ ONE STEP | |
| TASK | SDA+SURGE | FLOWDAS+SURGE | SDA+SURGE | FLOWDAS+SURGE |
|---|---|---|---|---|
| LORENZ | 82.9% | 81.5% | 75.2% | 71.3% |
| NS-SR | 59.2% | 50.8% | 47.2% | 18.6% |
| NS-SO | 59.9% | 51.4% | 40.2% | 19.5% |
| WEATHER | 51.7% | 85.9% | 30.1% | 45.3% |

*Table 8.* Ablation on the role of observation guidance ($N = 50$). Replacing FlowDAS with a plain diffusion model (DM) that receives no observation guidance leads to a notable drop in ESS, supporting the view that guidance is the primary driver of effective sample size.

| BACKBONE | TASK | MEAN ESS/$K$ ↑ |
|---|---|---|
| DM | NAVIER-STOKES | 7.7% |
| FLOWDAS | NAVIER-STOKES | 20.9% |
| DM | WEATHER | 19.7% |
| FLOWDAS | WEATHER | 51.7% |

benchmarks. We are working with this problem and trying to develop an interpretable ensemble method for better final output, as demonstrated in Figure 13.

**ESS analysis.** A common concern with particle-filter methods is weight degeneracy: a single particle dominates the ensemble and the effective sample size (ESS) collapses to one. Table 7 shows the particle system stays well-distributed. ESS drops as $N$ grows to 100, as expected for particle filters, but stays well above 1% on every task. The guided proposal avoids full weight collapse. ESS depends on the quality of the proposal, and in SURGE the proposal is shaped by observation guidance. Stochasticity in the diffusion sampler spreads particles, but guidance pulls them back toward high-likelihood regions, so the post-guidance weights stay closer to uniform. The ESS we report reflects this alignment with the posterior, not low observation noise. On harder tasks, SURGE becomes more selective and drops low-weight particles through resampling.

To check this, we run an ablation on Navier-Stokes and Weather ($N = 50$, on a subset), replacing FlowDAS with a plain diffusion model that receives no observation guidance (Table 8). ESS drops sharply (NS: $20.9\% \to 7.7\%$; Weather: $51.7\% \to 19.7\%$).

**Ablation Study.** To further investigate the efficacy of the proposed SURGE framework, we conducted ablation study. We firstly focus on the Lorenz system experiment to demonstrate the results with maximum clarity. This choice is motivated by the highly distinct nature of chaotic Lorenz trajectories, which offers the most discriminative visualization for analyzing the specific contributions of each component. We decompose the weight computation in the SURGE particle filter into two essential components to ensure accurate posterior estimation. The first is the Reward Term, defined as $(t + \Delta t)R(X^i_{t+\Delta t}) - tR(X^i_t)$, which quantifies the incremental gain in data fidelity by evaluating the alignment between the predicted state and observations via pixel-level likelihoods. The second is the Guidance Term, expressed as $-V(t)\nabla_x G(X^i_t|x_t) \cdot \sqrt{\Delta t}\xi^i_t - \frac{1}{2}V(t)^2\|\nabla_x G(X^i_t|x_t)\|^2\Delta t$, which functions as a Girsanov correction factor that mathematically compensates for the bias introduced by the external guidance drift, ensuring that the particle weights correctly reflect the approximation-free target posterior distribution. The impact of individual components is visualized in the ablation study. Figure 18 and Figure 19 display the predicted trajectories upon removing the guidance and reward terms, respectively. Notably, removing all SURGE features reverts the performance to the FlowDAS baseline, as shown in Figure 20. Collectively, these results demonstrate the importance of both terms for accurate trajectory estimation. The quantitative result over evaluation data is in Table 9, shows both terms are essential for SURGE accurate prediction.

## B.6. Computational Resources

All experiments were conducted on a single NVIDIA RTX PRO 6000 with 96GB memory. Training the diffusion model required approximately 1.5 hours for Lorenz system and 2 hours for Navier-stokes flow. We utilize pretrained model form

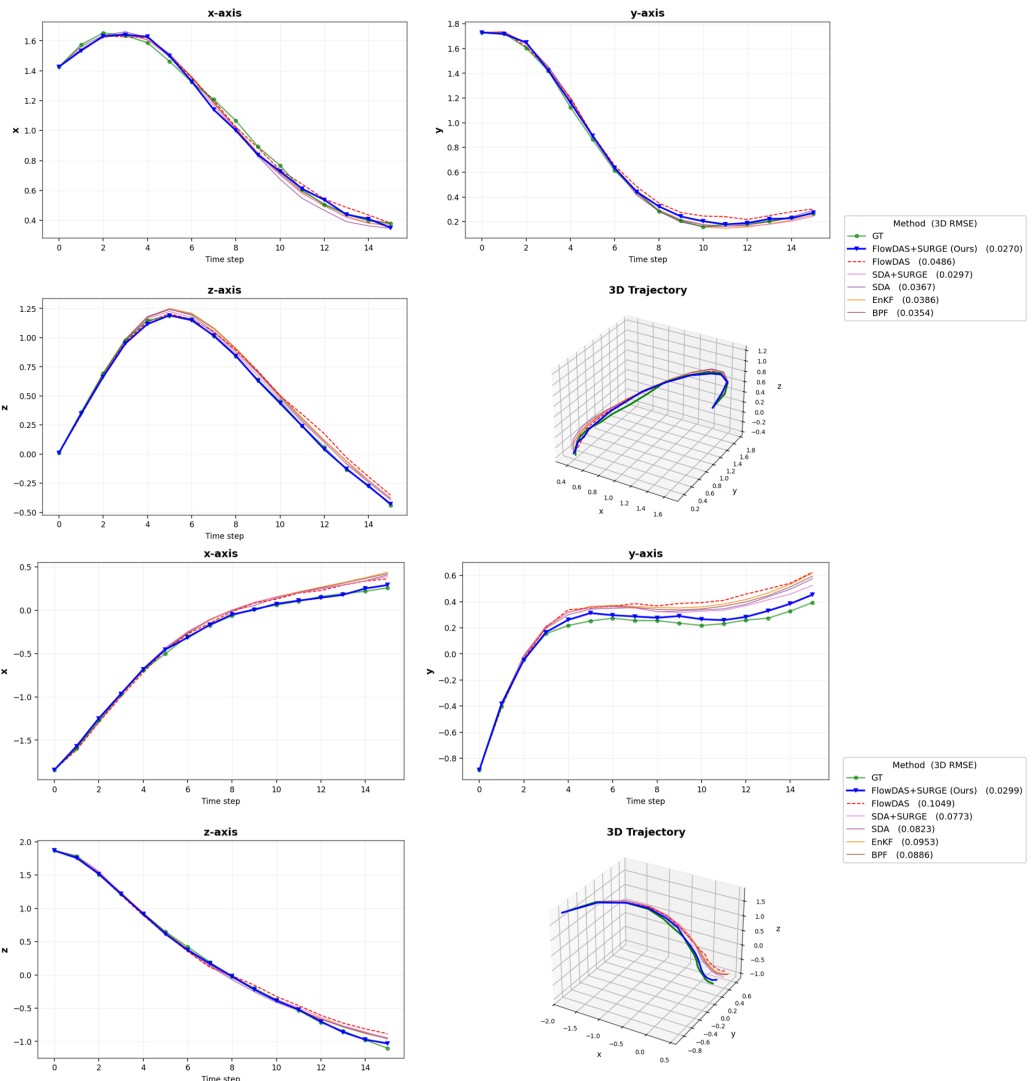

*Figure 7.* More trajectory wise comparison between baselines and SURGE on Lorenz system. The blue SURGE's trajectory align closer to green real trajectory compare with baselines' result.

FlowDAS in weather forecasting task. Inference with the SURGE filter took approximately 30 seconds per time step for Lorenz system, 0.5 seconds per time step for Navier-stokes and 100 seconds per time step based one the same setting with FlowDAS.

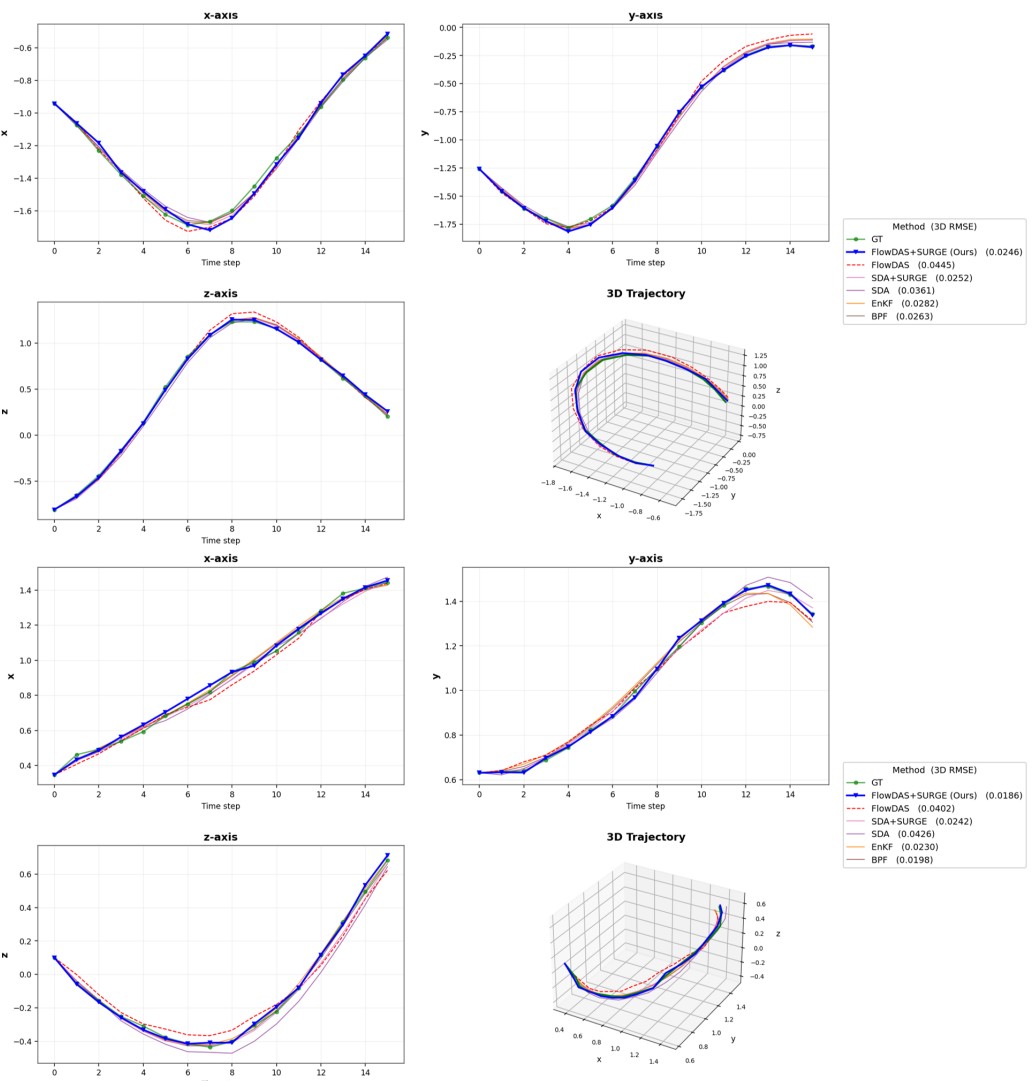

*Figure 8.* More trajectory wise comparison between baselines and SURGE on Lorenz system. The blue SURGE's trajectory align closer to green real trajectory compare with baselines' result.

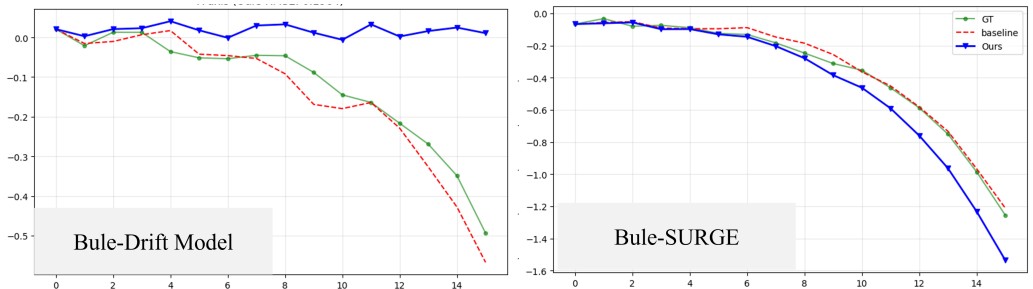

*Figure 9.* Failure case analysis of the Lorenz system under partial observation. When the trajectory is initialized near the null-isocline ($x \approx 0$), the drift model exhibits significant oscillatory instability (Left). In this scenario, although the SURGE guidance attempts to correct the bias, the poor quality of the underlying proposal distribution leads to excessive concentration of particle weights and trajectory overshooting (Right), highlighting the algorithm's dependence on the stability of the surrogate model.

Drift Model (Diffusion Surrogate) Prediction

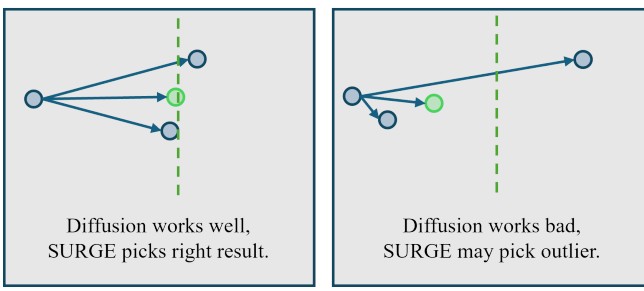

*Figure 10.* Illustration of `SURGE` behavior under unstable and erroneous predictions from the diffusion surrogate, and the resulting particle degeneracy.

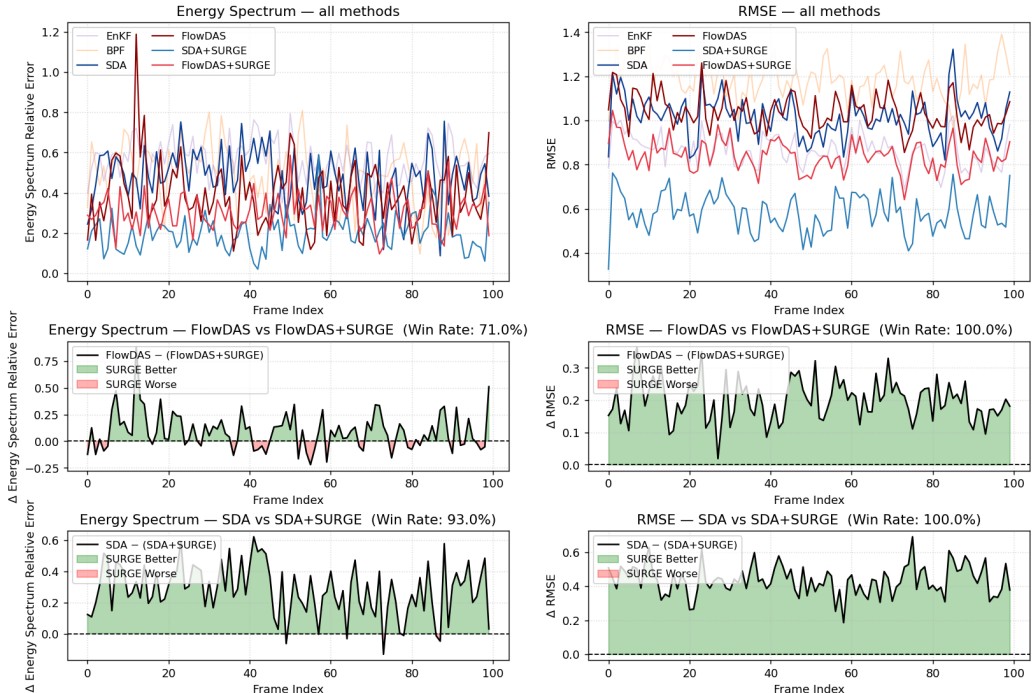

*Figure 11.* More trajectory wise comparison between baselines and `SURGE` on Navier-stokes flow in terms of Energy Spectrum Relative Error and RMSE, demonstrating that `SURGE` consistently outperforms all baselines.

*Table 9.* Quantitative result in ablation study of Lorenz system prediction task.

| METHOD | RMSE ↓ | $W_1$ ↓ |
|---|---|---|
| FLOWDAS W/O DA | 0.0666 | 0.0479 |
| FLOWDAS | 0.0545 | 0.0388 |
| **SURGE** | **0.0502** | **0.0363** |
| SURGE W/O REWARD | 0.0647 | 0.0486 |
| SURGE W/O GUIDANCE | 0.0608 | 0.0428 |

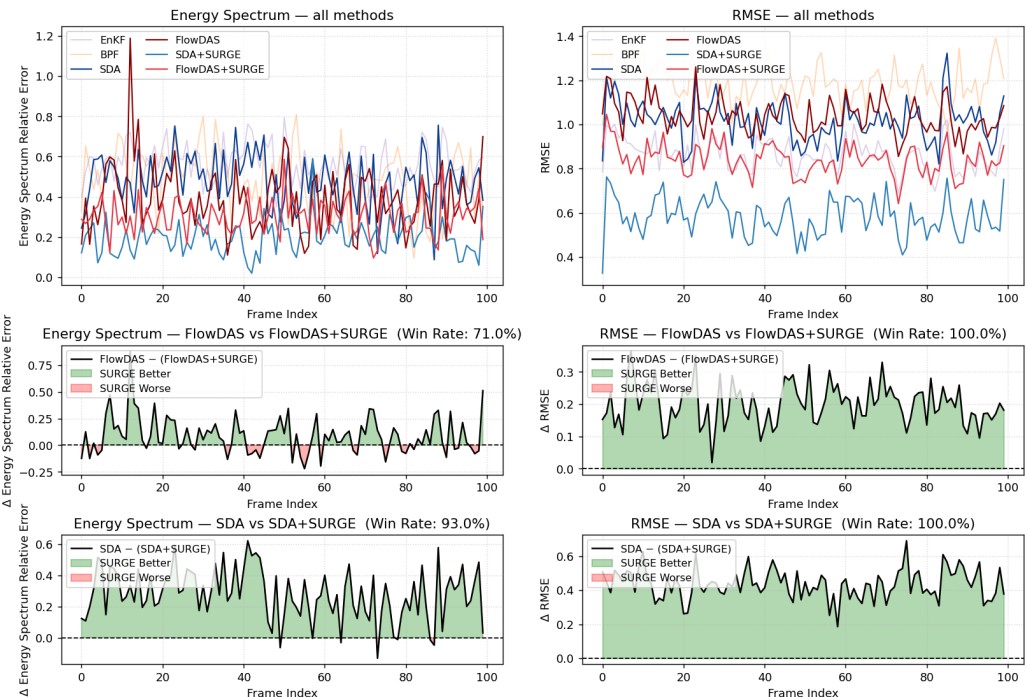

*Figure 12.* More trajectory wise comparison between baselines and SURGE on Navier-stokes flow in terms of Energy Spectrum Relative Error and RMSE, demonstrating that SURGE consistently outperforms all baselines.

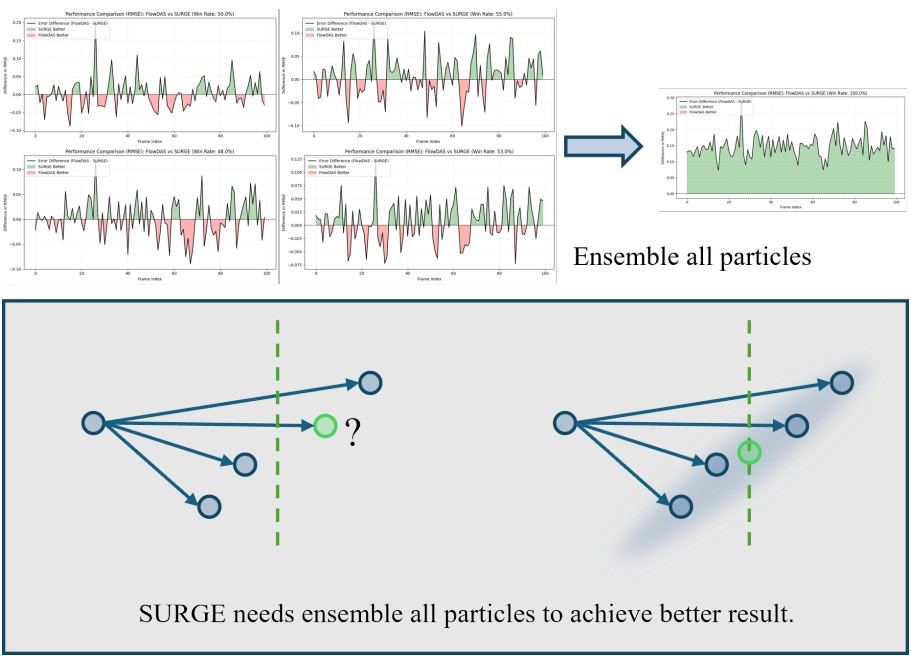

*Figure 13.* Impact of Ensemble Averaging. Individual particles (left) exhibit high stochastic variance, whereas the ensemble mean (right) effectively outperform in RMSE. This shows that SURGE relies on all particles for robust estimation.

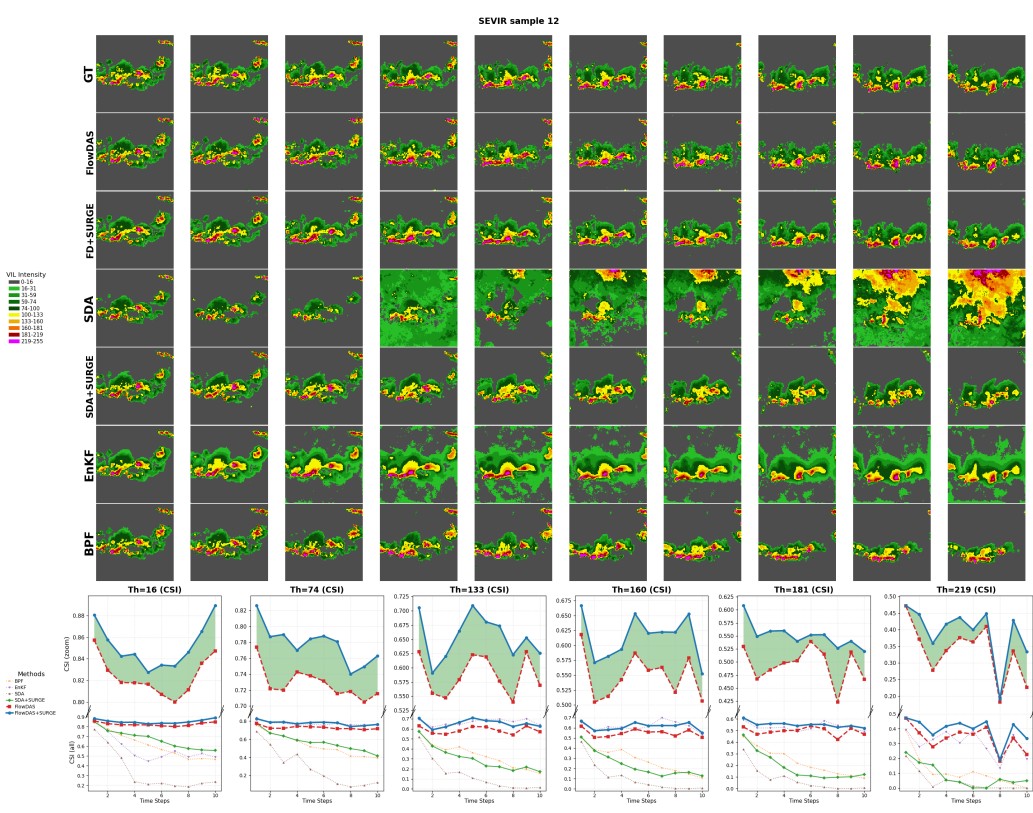

*Figure 14.* More trajectory wise comparison between baselines and SURGE on weather forecasting in terms of VIL intensity visualization and Critical Success Index (CSI), demonstrating that SURGE consistently outperforms baselines.

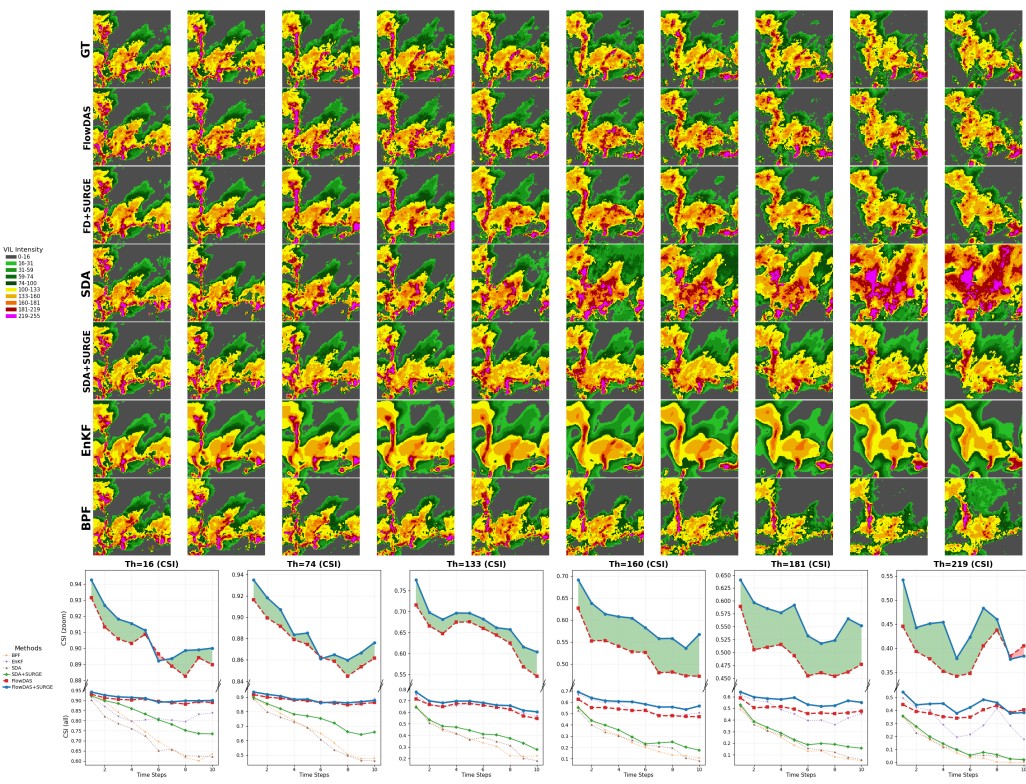

*Figure 15.* More trajectory wise comparison between baselines and SURGE on weather forecasting in terms of VIL intensity visualization and Critical Success Index (CSI), demonstrating that SURGE consistently outperforms baselines.

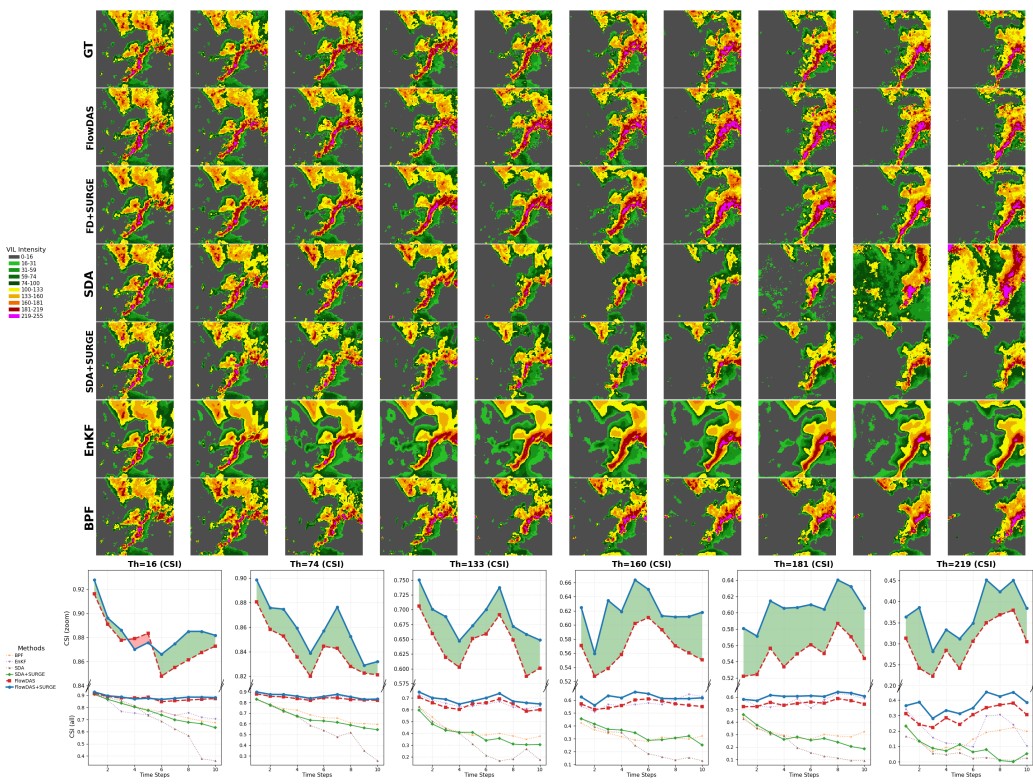

*Figure 16.* More trajectory wise comparison between baselines and SURGE on weather forecasting in terms of VIL intensity visualization and Critical Success Index (CSI), demonstrating that SURGE consistently outperforms baselines.

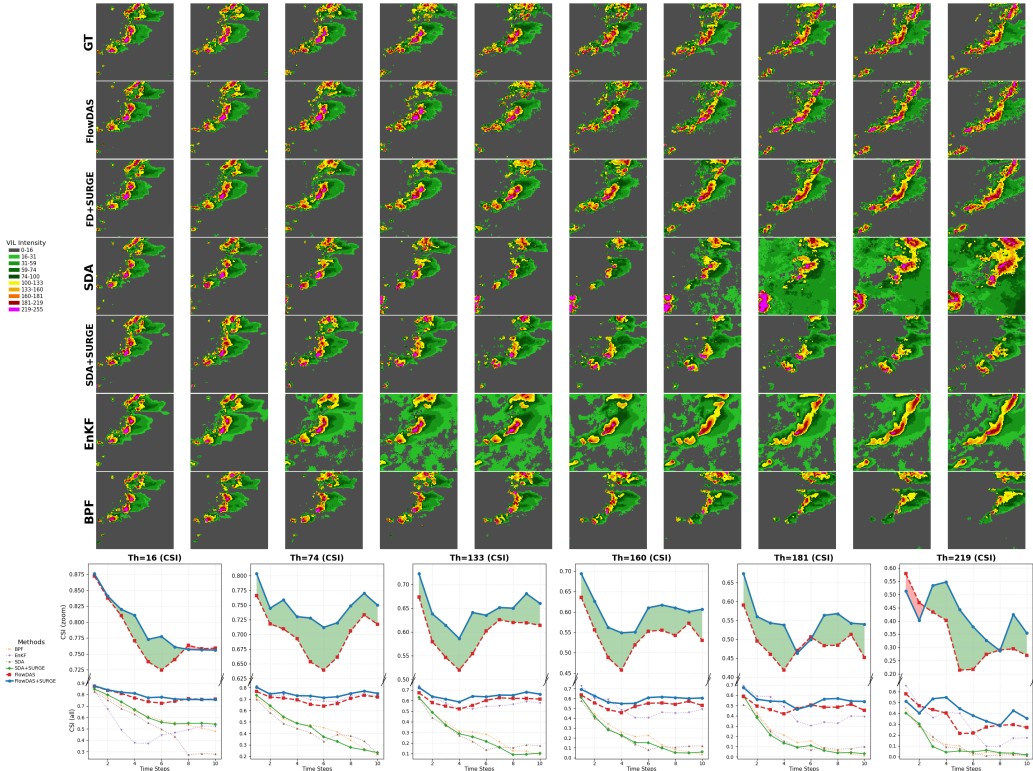

*Figure 17.* More trajectory wise comparison between baselines and `SURGE` on weather forecasting in terms of VIL intensity visualization and Critical Success Index (CSI), demonstrating that `SURGE` consistently outperforms baselines.

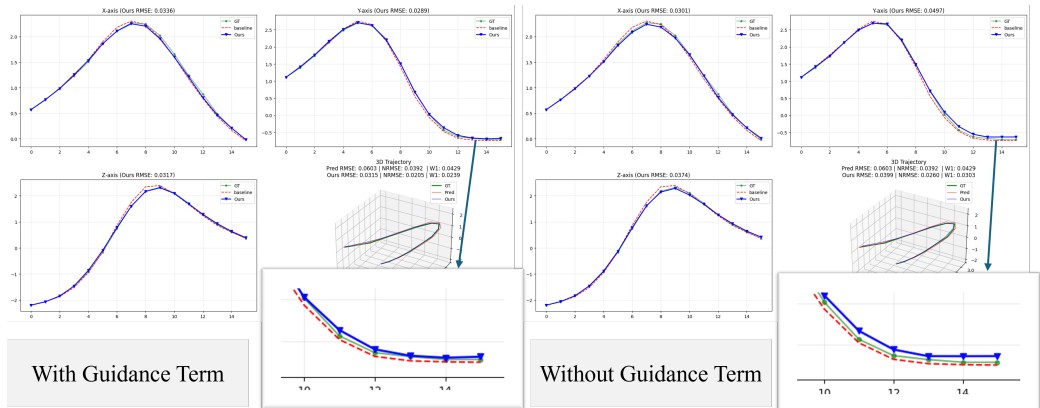

*Figure 18.* Ablation of the guidance term. Without guidance (right), the trajectory fails to correct drift and degrades to the baseline, highlighting the term's necessity for accurate tracking (left).

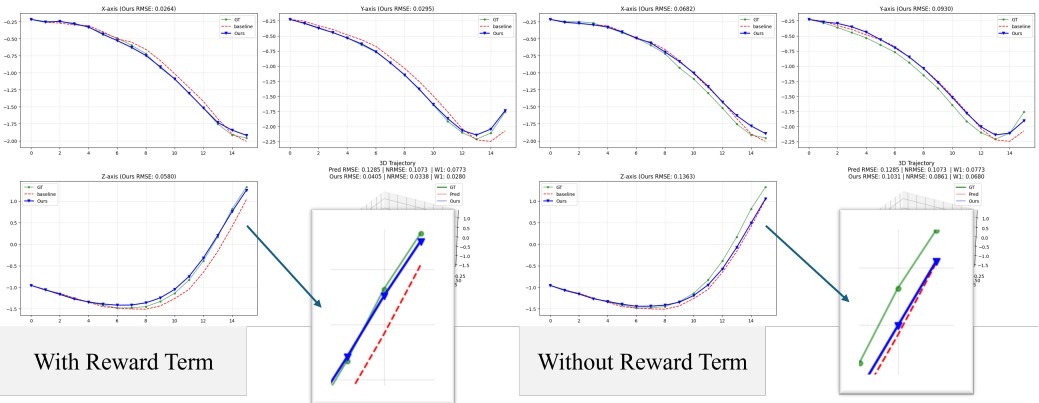

*Figure 19.* Ablation of the reward term. Without reward (right), the trajectory fails to correct drift and degrades to the baseline, highlighting the term's necessity for accurate tracking (left).

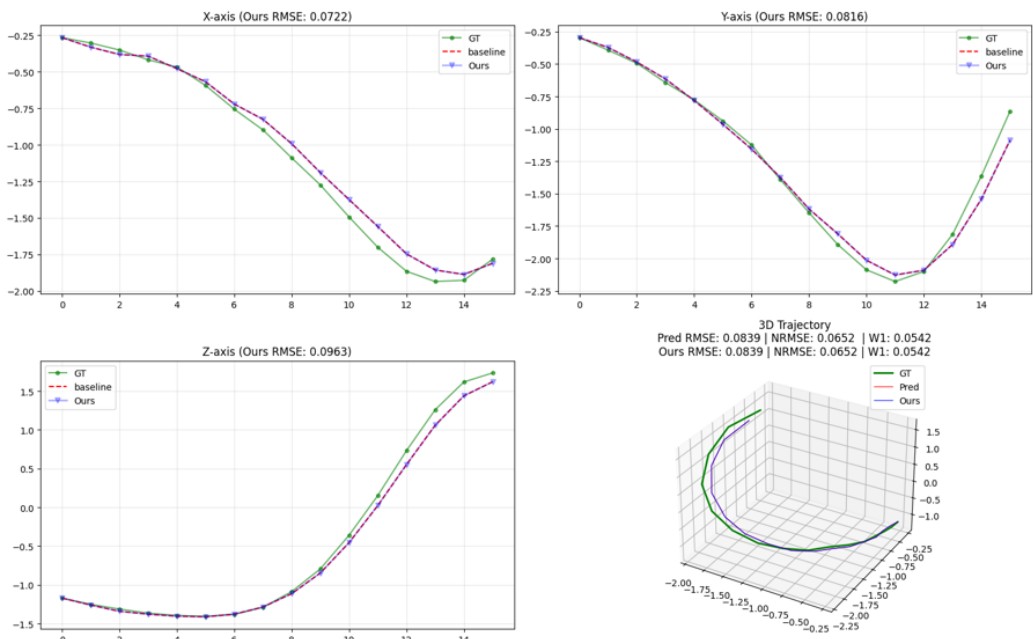

*Figure 20.* Ablation of SURGE weight computing and resampling. The trajectory is same as FlowDAS predicted.

