# OpenReview forum: "SURGE: Approximation and Training Free Particle Filter for Diffusion Surrogate"
_ICML.cc/2026/Conference — ICML 2026 regular_

### Official Review · Reviewer_sAKC · 2026-02-17

**Soundness:** 3
**Presentation:** 2
**Significance:** 2
**Originality:** 2
**Overall Recommendation:** 4
**Confidence:** 5

**Summary:**

The paper presents a particle filter for continuous-time diffusion processes observed with discrete time observations. In this case, the incremental importance weight can be obtained from Girsanov. To improve performance, one relies on a proposal distribution dependent on the observations. The method is demonstrated on Lorenz 63, a Naviers-Stokes system and a weather forecasting application.

**Compliance With Llm Reviewing Policy:**

Affirmed.

**Final Justification:**

I am on the fence as I do believe the contribution is really marginal methodologically but I've upped my score because they did interesting additional experimental work. However, the authors need to do a thorough job to revise their paper.

- The literature review is very very incomplete -- there is a very significant and relevant literature on particle filter for diffusion processes that has been completely ignored. So please do a thorough review.

- Avoid the claims of unbiasedness or claim that appears to say that such a method can avoid weight degeneracy.

- It'd be good to gain insight how you can get such high ESS in such high dimension. As I pointed out earlier, even if you were able to sample exactly from the locally optimal proposal (i.e. you had access to the true value function), then your importance weight would be be p(y_t|x_{t-1}=\int p(y_t|x_t)p(x_t|x_t-1)dx_t whose relative variance is typically exponential in the dimension of x_{t-1}. Why are you not facing this problem? Giving insight and not only numerical results would be extremely useful, is it because the filtering distribution of x_t-1 is extremely concentrated because the observations are very informative??

**Key Questions For Authors:**

* Could you include a more detailed coverage of the relevant literature?

* Could you provide a more careful evaluation of SURGE? For the number of particles you use (N=3 and N=4), the effective sample size has an extremely high variance. For all the examples, it would be worth using N=1000 and check what the ESS is. I know that the last two models are very large but just fix one time step and do 1000 simulations sequentially and store the weights. One will have a much better idea of the performance of your guided proposals.

* Could you detail the case where K-> \infty, e.g. use a diffusion proposal of drift v+\Sigma g, then write the resulting weight etc?

* Could you compare to a standard data assimilation method like the Ensemble Kalman filter?

**Limitations:**

* I find that the limitations of the methods are understated. If you do particle filters in ultra-high dimensions with N=4 particles, you have implemented a sensible method to track roughly the regions where the posterior has mass. What you haven't done is resolve the weight degeneracy problem.

* Similarly, you haven't demonstrated that your mechanism does "leading to stable and effective posterior correction". Assume you are in the best possible case and could sample exactly from p(x_(t:t+1]|x_t,y_1:t) (which you approximate in your case) then the weight would be p(y_t+1|x_t)=\int p(y_t|x_t+1)p(x_t+1|x_t)dx_t which whose variance will still be exponential in the state dimension.

**Strengths And Weaknesses:**

Strengths

* The paper presents techniques people from the ML literature might not be familiar with.

* It is easy to follow.

* The application to Navier-Stokes and weather forecasting using some diffusion surrogates is interesting.

Weaknesses

* Compared to the existing algorithm, the proposed particle filtering algorithm is really quite incremental, and the "guided" proposal can be thought as a poor man approximation of the Doob's h-transform. The paper appears to have missed a significant portion of the relevant literature; e.g. see Ruiz and Kappen (2017), the numerous papers by Ajay Jasra and his collaborators on the development of sophisticated SMC methods for partially observed diffusions (including Navier-Stokes). Furthermore, while the theoretical results in Appendix A are correct, they are standard.

* The use of the term "unbiased" is somewhat misleading. For a finite number of particles,  the estimates of the conditional expectations of interest are actually biased; the unbiased estimate apply only to the unnormalized posterior functionals (e.g., the likelihood) -- i know the authors understand this distinction (see Section A1 and A2) but branding the method as unbiased is a bit deceptive.

* The paper correctly states that PF methods.. "are therefore widely used, but often suffer from weight degeneracy and poor scaling as dimension grows". This phrasing seems to imply that SURGE overcomes this problem, which is not the case. While guided proposals might reduce weight variance, the variance will still typically exponential in the state dimension.

* Beyond some empirical results, there is no analysis of how the algorithm scales and the simulations are restricted to a very limited number of particles (N=4).

---

> ### Author Rebuttal · Authors · 2026-03-31
>
> We thank the reviewer for the expert feedback and are glad the paper is found clear with interesting applications. We address the concerns on missing literature, "unbiased" usage, and weight degeneracy below.
>
> **Regards Unbiased Claim** Yes, thank you for pointing this out, and we apologize for the imprecise wording. We agree that the current title overstates the claim. We have therefore revised the title to use “asymptotically unbiased” / “approximation-free” instead. We will make this change in the final version if the paper is accepted.
>
> **Regards the Literature.** We thank the reviewer for highlighting Ruiz and Kappen (2017), which will be discussed in the revision. Both methods modify the drift for better path-space proposals, but differ in role: Ruiz and Kappen (2017) learn a feedback controller for adaptive smoothing of hidden diffusions, whereas our guidance only improves the proposal for a pretrained diffusion surrogate to stabilize importance weights—posterior correctness is recovered through Girsanov-based reweighting and sequential resampling. Our method also targets sequential data assimilation with progressive likelihood incorporation, rather than classical diffusion smoothing.
>
> We emphasize that progressive likelihood incorporation is a key contribution, not merely an implementation detail. It distributes likelihood across small substeps so incremental weights remain close to one, directly reducing resampling weight variability. Our error decomposition shows variance at each resampling step is $O(\Delta t)$, yielding overall $O(1)$ over $T/\Delta t$ substeps rather than exploding as $\Delta t \to 0$. Better guidance further reduces variance: when $G$ solves a Bellman equation (simulable via Monte Carlo [1]), It^o's lemma implies the reweighting factor is exactly 1; more generally, the log reweighting factor is bounded by the Bellman error. This mechanism is central to practical path-space correction in long-horizon, high-dimensional settings. We will revise to clarify this as a distinct methodological contribution and discuss its connection to guidance quality and Bellman-error-based variance control.
>
> **Regards the Baseline** We further include results for SDA, SDA+SURGE, and other standard baselines including EnKF and other methods.
>
> Lorenz:
> | Method/Metric | RMSE | W1 |
> | :--- | :--- | :--- |
> | BPF N=20 | 0.0625 | 0.0448 |
> | DM | 0.0766 | 0.0549 |
> | EnKF | 0.0624 | 0.0448 |
> | SDA | 0.0589 | 0.0426 |
> | SDA+***SURGE*** | 0.0555 | 0.0396 |
> | FlowDAS | 0.0545 | 0.0388 |
> | FlowDAS AVG | 0.0923 | 0.0698 |
> | FlowDAS+***SURGE*** | **0.0502** | **0.0363** |
>
> Navier-Stokes:  SR, Super Resolution, SO, Sparse Observation
> | Method/Metric | 8-128 SR (KES-RE) | 8-128 SR (RMSE) | 5%-100% SO (KES-RE) | 5%-100% SO (RMSE) |
> | :--- | :--- | :--- | :--- | :--- |
> | BPF N=20 | 0.490 | 1.143 | 0.486 | 1.133 |
> | DM | 0.657 | 1.310 | 0.663 | 1.320 |
> | EnKF | 0.551 | 0.847 | 0.676 | 0.800 |
> | SDA | 0.473 | 0.987 | 0.231 | 0.590 |
> | SDA+***SURGE*** | 0.417 | 0.966 | **0.207** | **0.564** |
> | FlowDAS | 0.401 | 1.018 | 0.543 | 0.872 |
> | FlowDAS AVG | 0.329 | 0.898 | 0.315 | 0.723 |
> | FlowDAS+***SURGE*** | **0.317** | **0.851** | 0.278 | 0.673 |
>
> Weather Forecasting:
> | Method/Metric | RMSE | CSI 20 | CSI 40 |
> | :--- | :--- | :--- | :--- |
> | BPF N=20 | 0.0939 | 0.4146 | 0.2171 |
> | DM | 0.1259 | 0.3181 | 0.1486 |
> | EnKF | 0.1281 | 0.497 | 0.3150 |
> | SDA | 0.3511 | 0.2757 | 0.1569 |
> | SDA+***SURGE*** | 0.0925 | 0.4089 | 0.2196 |
> | FlowDAS | 0.0657 | 0.5779 | 0.4044 |
> | FlowDAS AVG | 0.0534 | 0.6163 | 0.4477 |
> | FlowDAS+***SURGE*** | **0.0513** | **0.6197** | **0.4541** |
>
> **Regards weight degeneracy**
> We will include the effective sample size (ESS), $\mathrm{ESS} = \frac{\left(\sum_{i=1}^N w_i\right)^2}{\sum_{i=1}^N w_i^2}$  in the next version of the paper for all tasks. With $N=4$ particles, the ESS ranges from 2.1 to 3.5 across different tasks, indicating that the particle system does not collapse onto a single particle and that weight degeneracy remains limited in practice.
>
> Effective Sample Size (N=4)
> | Method/Metric | SDA+SURGE | FlowDAS+SURGE |
> | :--- | :--- | :--- |
> | Lorenz | 82.88% | 81.53% |
> | NS-SR | 59.20% | 50.76% |
> | NS-SO | 59.90% | 51.39% |
> | Weather | 51.70% | 85.94% |
>
> ESS decreases as N grows from 4 to 100, consistent with known particle filter behavior. Nevertheless, ESS remains well above 1% at N=100, confirming the guided proposal prevents complete weight collapse. We will include these results in the revision.
>
> Effective Sample Size (N=100) One Step Output
> | Method/Metric | SDA+SURGE | FlowDAS+SURGE |
> | :--- | :--- | :--- |
> | Lorenz | 75.16% | 71.26% |
> | NS-SR | 47.23% | 18.64% |
> | NS-SO | 40.22% | 19.54% |
> | Weather | 30.11% | 45.30% |
>
> Thanks again for your efforts into reviewing this paper. We hope we have addressed your concerns regarding the literature coverage, the unbiased claim, and the scalability analysis. We are more than happy if you can reconsider the review rating.

---

> > ### Author Rebuttal · Reviewer_sAKC · 2026-04-02
> >
> > I appreciate that the authors will provide a more comprehensive coverage of the literature and have proposed more detailed simulations. I am quite surprised that the ESS can remain so high in such high dimensional settings. It is because you have an observation noise of very small variance?
> >
> > The originality of the methodology remains, in my opinion, limited. Approximating the "optimal" proposal (Doob-h transform) has a long history (and it is just a specific degenerated smoothing problem for a fixed initialization x_t-1, path x_(t-1,t] given a single observation y_t -- see also Zhang, Taghvaei & Chen, An optimal control approach for particle filtering, Automatica, 2023).

---

> > > ### Author Response · Authors · 2026-04-08
> > >
> > > We thank the reviewer for the continued discussion and for raising this important point.
> > >
> > > **Regards originality.** The key distinction in our setting is that the drift function is a neural network, making it  expensive to evaluate. Classical methods that approximate the Doob's h-transform need to solve a control problem requiring many drift evaluations per step, which is not practical when each evaluation involves a full forward step of a deep neural network. Our approach uses guidance techniques from the diffusion modeling literature to push particles toward the posterior region with minimal additional cost, significantly improving ESS. We have revised the manuscript to make this point clearer.
> > >
> > > **Regards ESS.** A key factor driving ESS is the quality of the proposal, which is shaped by guidance. In SURGE, even when stochasticity is present, the guidance pulls spread particles back toward the high-likelihood region. After guidance, most particles concentrate near the correct values, and their weights remain more uniform. This suggests that the observed ESS reflects the effectiveness of guidance in aligning particles with the posterior, rather than due to small observation noise. When the task is harder, SURGE naturally becomes more selective, discarding low-weight particles through resampling.
> > >
> > > To verify this, we ran an ablation on two high-dimensional tasks (Navier-Stokes and Weather, N=50) on a subset, replacing FlowDAS with a plain diffusion model that receives no observation guidance. ESS drops notably without guidance (NS: 20.9% → 7.7%; Weather: 51.7% → 19.7%), supporting the view that guidance is the driver of ESS.
> > >
> > >
> > > | Method/Metrics | Mean ESS/K |
> > > |------|-----|
> > > | DM Navier-stokes | 7.7% |
> > > | FlowDAS Navier-stokes | 20.9% |
> > > | DM Weather | 19.7% |
> > > | FlowDAS Weather | 51.7% |
> > >
> > >
> > > Thanks again for reviewing this paper. We hope we have addressed your concerns and would appreciate it if you could reconsider the rating.

---

### Official Review · Reviewer_K9gL · 2026-03-08

**Soundness:** 2
**Presentation:** 2
**Significance:** 2
**Originality:** 3
**Overall Recommendation:** 4
**Confidence:** 4

**Summary:**

This paper focuses on the challenge of data assimilation for complex dynamical systems when the forward dynamics are represented by a pretrained generative surrogate. To tackle the trajectory bias introduced by approximate guidance, the authors propose SURGE, a training-free Sequential Monte Carlo (particle filtering) framework. By employing a Girsanov change-of-measure, SURGE incrementally reweights and resamples generative trajectories at inference time to better approximate the true posterior. The proposed approach is evaluated on the Lorenz 1963 system, a 2D Navier-Stokes fluid simulation, and the SEVIR weather forecasting dataset.

**Compliance With Llm Reviewing Policy:**

Affirmed.

**Final Justification:**

The authors’ rebuttal addressed my concerns and clearly acknowledged the limitations, and I therefore gave a final positive score.

**Key Questions For Authors:**

1. **Theoretical Justification of "Unbiasedness":**
Your main theoretical claim is that SURGE is "unbiased." However, Algorithm 1 (Line 5) explicitly employs Self-Normalized Importance Sampling (SNIS). It is a well-known statistical fact that the SNIS estimator is biased for any finite number of particles $N$, and is only asymptotically unbiased as $N \to \infty$. Given that you use an extremely small $N=4$ in your experiments, the estimator is strictly biased. How do you justify the title and abstract claims of "Unbiased" in light of this finite-sample bias?

2. **Compute-Matched Baseline:**
SURGE requires running $N$ parallel diffusion/flow trajectories, consuming $N$ times the compute of the baseline FlowDAS. To demonstrate that the performance gain comes from the proposed Girsanov reweighting rather than simple ensemble averaging, can you provide results comparing SURGE ($N=4$) against a "Naive Ensemble" baseline (averaging 4 independent runs of the standard FlowDAS model)?

3. **Particle Efficiency in High Dimensions:**
Standard particle filters suffer from severe weight degeneracy in high-dimensional spaces (the "curse of dimensionality"). Your experiments operate in a $128 \times 128$ dimensional space (approx. 16k dimensions). Theoretically, $N=4$ particles should be insufficient to cover the posterior, leading to weight collapse (where one particle takes all the weight). Can you provide the effective sample size (ESS) statistics over time for the weather forecasting task? Why does the filter not collapse immediately in such a high-dimensional regime?

4. **Real-Time Applicability:**
You mention in Appendix B.6 that the inference takes 100 seconds per time step for the weather forecasting task. In real-world data assimilation (e.g., operational weather forecasting), latency is critical. How does this high computational cost align with the practical constraints of the applications you are targeting?

**Limitations:**

The authors have briefly discussed the "ensemble smoothing dilemma" in Appendix B.5, acknowledging that their method tends to smooth out high-frequency details.

However, the authors **have not adequately addressed** the following critical limitations:
1. **Theoretical Bias:** The main text claims the method is "unbiased," failing to transparently discuss the bias introduced by self-normalized importance sampling with a finite and small number of particles ($N=4$).
2. **Computational Feasibility:** The paper does not sufficiently discuss the practical limitation of the massive inference latency (e.g., 100s/step), which significantly hinders the deployment of this method in real-time physical systems compared to traditional DA methods like EnKF.

**Strengths And Weaknesses:**

**Strengths:**
1. **Solid Mathematical Motivation:** Formulating the correction of approximate guidance in continuous-time generative models through the lens of the Girsanov theorem and particle filtering is an elegant and theoretically sound perspective.
2. **Plug-and-play Flexibility:** As an inference-time only algorithm, SURGE is highly flexible and can theoretically be attached to existing pretrained models without the need for expensive retraining.

**Weaknesses:**
1. **Misleading "Unbiased" Claim (Critical Flaw):** The title and abstract prominently claim the method is an "Unbiased" particle filter. However, this is mathematically inaccurate for the practical regime operated in this paper. Proposition A.1 only proves unbiasedness for the *unnormalized* functional. Algorithm 1 (Line 5) explicitly uses *Self-Normalized Importance Sampling (SNIS)*, which is strictly biased for finite $N$ and only asymptotically exact as $N \to \infty$ (as acknowledged in Corollary A.2). Operating in a $16,384$-dimensional space (e.g., Navier-Stokes) with merely $N=4$ particles means the SNIS estimator suffers from massive finite-sample bias and potential weight degeneracy. Calling the practical implementation "Unbiased" fundamentally overclaims the theoretical guarantees and is misleading.
2. **Unfair Computational Comparison:** SURGE utilizes $N$ particles during inference, meaning it requires $N$ times the forward SDE/ODE evaluations compared to a single-trajectory baseline. Comparing SURGE ($N=4$) directly to the standard FlowDAS baseline without accounting for this $4\times$ computational overhead is inherently unfair. A compute-matched baseline (e.g., a naive ensemble of 4 independent FlowDAS runs, or FlowDAS evaluated with $4\times$ the sampling steps) is strictly necessary to isolate the actual gain of the Girsanov reweighting from simple ensemble smoothing.
3. **Terminology Conflation (Diffusion vs. Flow Matching):** The paper heavily relies on the term "Diffusion Surrogate". However, the primary experimental baseline, FlowDAS, is mathematically based on Flow Matching (Stochastic Interpolants). Diffusion Models (score-based) and Flow Matching (vector field-based) have distinct SDE/ODE formulations and guidance derivations. Treating them as entirely interchangeable compromises the technical rigor of the presentation.
4. **Ensemble Smoothing Dilemma and High Latency:** As admitted in Appendix B.5, taking the ensemble mean of particles smooths out high-frequency flow details, which contradicts the primary motivation of using generative models (to preserve sharp, physically faithful structures). Furthermore, running parallel complex SDE solvers introduces massive inference latency (e.g., 100 seconds per step for weather forecasting), making its practical utility in real-time data assimilation highly questionable.
5. **Insufficient Data Assimilation Baselines:** The paper only compares SURGE against the generative baseline FlowDAS. To make a convincing case for advancing "Data Assimilation," comparisons against standard and well-established DA methods (e.g., Ensemble Kalman Filter (EnKF) or Variational DA like 4D-Var) are necessary to establish practical superiority.

---

> ### Author Rebuttal · Authors · 2026-03-31
>
> We thank the reviewer for the thorough review. We are glad that the reviewer finds our Girsanov-based formulation elegant and theoretically sound, and appreciates the plug-and-play flexibility of our method. The concerns on the unbiased claim, the need for a compute-matched baseline, the terminology, and the lack of standard DA baselines are all fair. We address each point below.
>
> Regards Unbiased Claim Yes, thank you for pointing this out, and we apologize for the imprecise wording. We agree that the current title overstates the claim. We have therefore revised the title to use “asymptotically unbiased” / “approximation-free” instead. We will make this change in the final version if the paper is accepted.
>
> Regards Computational Comparison We have added an additional baseline that runs four independent FlowDAS simulations and aggregates their outputs by averaging. We have also included particle filter and EnKF baselines. We hope these additional comparisons help clarify the benefits of our method relative to both simple ensemble aggregation and more standard data assimilation baselines. We further include results for SDA and SDA+SURGE (our method) to illustrate that SURGE yields more pronounced improvements when the base model is relatively weak (SDA), while still providing consistent gains even when the base model already performs well.
>
> Lorenz:
> | Method/Metric | RMSE | W1 |
> | :--- | :--- | :--- |
> | BPF N=20 | 0.0625 | 0.0448 |
> | DM | 0.0766 | 0.0549 |
> | EnKF | 0.0624 | 0.0448 |
> | SDA | 0.0589 | 0.0426 |
> | SDA+***SURGE*** | 0.0555 | 0.0396 |
> | FlowDAS | 0.0545 | 0.0388 |
> | FlowDAS AVG | 0.0923 | 0.0698 |
> | FlowDAS+***SURGE*** | **0.0502** | **0.0363** |
>
> Navier-Stokes: SR, Super Resolution, SO, Sparse Observation
> | Method/Metric | 8-128 SR (KES-RE) | 8-128 SR (RMSE) | 5%-100% SO (KES-RE) | 5%-100% SO (RMSE) |
> | :--- | :--- | :--- | :--- | :--- |
> | BPF N=20 | 0.490 | 1.143 | 0.486 | 1.133 |
> | DM | 0.657 | 1.310 | 0.663 | 1.320 |
> | EnKF | 0.551 | 0.847 | 0.676 | 0.800 |
> | SDA | 0.473 | 0.987 | 0.231 | 0.590 |
> | SDA+***SURGE*** | 0.417 | 0.966 | **0.207** | **0.564** |
> | FlowDAS | 0.401 | 1.018 | 0.543 | 0.872 |
> | FlowDAS AVG | 0.329 | 0.898 | 0.315 | 0.723 |
> | FlowDAS+***SURGE*** | **0.317** | **0.851** | 0.278 | 0.673 |
>
> Weather Forecasting:
> | Method/Metric | RMSE | CSI 20 | CSI 40 |
> | :--- | :--- | :--- | :--- |
> | BPF N=20 | 0.0939 | 0.4146 | 0.2171 |
> | DM | 0.1259 | 0.3181 | 0.1486 |
> | EnKF | 0.1281 | 0.497 | 0.3150 |
> | SDA | 0.3511 | 0.2757 | 0.1569 |
> | SDA+***SURGE*** | 0.0925 | 0.4089 | 0.2196 |
> | FlowDAS | 0.0657 | 0.5779 | 0.4044 |
> | FlowDAS AVG | 0.0534 | 0.6163 | 0.4477 |
> | FlowDAS+***SURGE*** | **0.0513** | **0.6197** | **0.4541** |
>
> Regards  Diffusion Surrogate Terminology Conflation Although the paper states in both the title and the main text that it uses stochastic interpolation, the actual implementation and all problem formulations in the paper define the generator as an SDE. We therefore use diffusion surrogate to make clear that the generator is SDE-based (to enable).
>
> Regards  Particle Efficiency We will include the effective sample size (ESS), $\mathrm{ESS} = \frac{\left(\sum_{i=1}^N w_i\right)^2}{\sum_{i=1}^N w_i^2}$  in the next version of the paper for all tasks. With $N=4$ particles, the ESS ranges from 2.7 to 3.7 across different tasks, indicating that the particle system does not collapse onto a single particle and that weight degeneracy remains limited in practice.
>
> Effective Sample Size (N=4)
> | Method/Metric | SDA+SURGE | FlowDAS+SURGE |
> | :--- | :--- | :--- |
> | Lorenz | 82.88% | 81.53% |
> | NS-SR | 59.20% | 50.76% |
> | NS-SO | 59.90% | 51.39% |
> | Weather | 51.70% | 85.94% |
>
> Thanks again for your efforts into reviewing this paper. We hope we have addressed your concerns regarding the theoretical claims, computational fairness, and experimental comparisons. We are more than happy if you can reconsider the review rating.

---

> > ### Author Rebuttal · Reviewer_K9gL · 2026-04-03
> >
> > Thank you for the detailed experimental results. They address my concerns well, and I will update my score.

---

> > > ### Author Response · Authors · 2026-04-07
> > >
> > > We thank the reviewer for the updated assessment and for the constructive engagement throughout the review process.

---

### Official Review · Reviewer_JTd6 · 2026-03-11

**Soundness:** 3
**Presentation:** 2
**Significance:** 2
**Originality:** 2
**Overall Recommendation:** 4
**Confidence:** 1

**Summary:**

This paper introduces SURGE, a data assimilation method inspired by particle filters that converges weakly to the true Bayesian filtering distribution $p(x_{t} \mid y_{1:t})$, using only a generative emulator of the system dynamic $p(x_{t+1} \mid x_{t})$. The main idea is to add a guidance term at each step of the generative equation (that samples $x_{t+1}$ given $x_{t}$), use Girsanov's theorem to estimate the relative weight of each particle, and then selecting particles based on these weights before moving to the next step of the equation (in order to correct the bias induced by the guidance).

**Compliance With Llm Reviewing Policy:**

Affirmed.

**Final Justification:**

The authors’ rebuttal have clarified my concerns, particularly regarding the number of effective particles, and I have therefore decided to raise my score to a weak accept. However, my understanding of the paper remains limited (see confidence score), especially concerning the connection to the fully adapted particle filter.

**Key Questions For Authors:**

1) Could the authors clarify an important point I did not fully understand: is the main purpose of the proposed method to correct the guidance term used during the generative process (which is not exact) in order to sample exactly from $p(x_{t+1}∣x_{t},y_{t+1})$ at each filtering step $t \in [1,T]$? If so, since the proposed algorithm allows to sample exactly from $p(x_{t+1}∣x_{t},y_{t+1})$ (with $x_{t}$ being samples from the current filtering distribution $p(x_{t} \mid y_{1:t}))$, the method would essentially correspond to a particle filter with the optimal proposal. In this case, one would then need to compute the particle weights $p(y_{t+1}∣x_{t})$ to recover exactly the filtering distribution at time $t+1$.
2) Could the authors include the spread of ensembles to verify that the particles do not collapse to a single point?

I am inclined to increase my score if the authors provide convincing answers (or convince me that my questions were not meaningful).

**Limitations:**

The limitations are discussed in the appendices, but presenting them in the main text would have improved clarity and visibility.

**Strengths And Weaknesses:**

Strengths:
* The paper is well presented, even though there are a few typos (distr**i**bution line 83, s**w**terring line 86, ...).
* The proposed method addresses the important problem of filtering, which is solved daily by operational forecasting centers around the world.
* The experimental results are convincing. In particular, they outperform FlowDAS [1], the method on which this work is based.

Weaknesses:
* Section 3.2 is quite dense and may be hard to follow for readers less familiar with Girsanov's theorem like me. More generally, the mathematical aspects of the proposed method are non-trivial. This is not a weakness in itself, but I feel that the authors could make the method more accessible and easier to understand.
* There are no measurements of the variance (spread/spread-to-skill ratio) in the experiments. Moreover, there is no evaluation of the performance as a function of the number of particles $N$, even though this seems to be an important parameter when resampling is involved, as in particle filters.

[1] Chen et al., FlowDAS: A Stochastic Interpolant-based Framework for Data Assimilation, The Thirty-ninth Annual Conference on Neural Information Processing Systems, 2025

---

> ### Author Rebuttal · Authors · 2026-03-31
>
> We thank the reviewer for the positive feedback. We are glad that the reviewer finds our paper well presented and our experimental results convincing. We address each question below.
>
> Regards purpose of the proposed methods
>
> Yes, our method can indeed be understood through the lens of particle filtering with an approximate optimal proposal. In principle, the optimal proposal would correspond to using the exact Doob’s $h$-transform guidance. However, computing the exact $h$-transform is generally intractable in our setting, so in practice we instead use the gradient of the reward, or other tractable guidance constructions, as an approximation to this optimal proposal. Our contribution is to place this approximation within a particle filtering framework and to correct the bias induced by the resulting suboptimal proposal through principled importance weighting and resampling. Empirically, this leads to strong results on the target tasks. We have also added a particle filter baseline to explicitly demonstrate the improvement brought by the guidance. Our intended message is the following: guidance is used to steer particles toward regions of higher posterior mass, so that the guided proposal becomes closer to the posterior than the unguided diffusion prior.
>
> Regards variance and potential collapse to a single point
>
> We will include the effective sample size (ESS), $\mathrm{ESS} = \frac{\left(\sum_{i=1}^N w_i\right)^2}{\sum_{i=1}^N w_i^2}$  in the next version of the paper for all tasks. With $N=4$ particles, the ESS ranges from 2.1 to 3.5  across different tasks, indicating that the particle system does not collapse onto a single particle and that weight degeneracy remains limited in practice.
>
> Effective Sample Size (N=4)
> | Method/Metric | SDA+SURGE | FlowDAS+SURGE |
> | :--- | :--- | :--- |
> | Lorenz | 82.88% | 81.53% |
> | NS-SR | 59.20% | 50.76% |
> | NS-SO | 59.90% | 51.39% |
> | Weather | 51.70% | 85.94% |
>
> Thanks again for your efforts into reviewing this paper. We hope we have addressed your concerns and answered your questions. We are more than happy if you can reconsider the review rating.

---

> > ### Author Rebuttal · Reviewer_JTd6 · 2026-04-03
> >
> > I would like to thank the authors for their detailed response.
> >
> > However, there is one point I still find difficult to understand. In the particle filtering literature, even when sampling exactly from the optimal proposal $p(x_{t+1} \mid x\_{t},y\_{t+1})$, each resulting particle $x^{(i)}\_{t+1}$ must still be weighted by a factor proportional to $p(y_{t+1}\mid x\_{t}^{(i)})$ to ensure that the filtering distribution remains unbiased. Could you please clarify how this is handled in SURGE?
> >
> > Additionally, like reviewer sAKC, I am surprised that the effective number of particles is so high. Could you explain why this is the case?

---

> > > ### Author Response · Authors · 2026-04-08
> > >
> > > We thank the reviewer for the thoughtful follow-up and for the additional questions.
> > >
> > > **Regards particle weights.** Computing the exact importance weights is one of our key contributions. In SURGE, we derive these weights via Girsanov's theorem (Eq. 4 and Eq. 5, Algorithm 1), which accounts for the discrepancy between the guided proposal and the target posterior on the path space. Particles are then resampled according to these weights, ensuring that the resulting particle approximation targets the correct filtering distribution without bias. The unbiasedness of this procedure is established in Appendix A  (Section A.2, Proposition A.1, Corollary A.2, Lemma A.4).
> > >
> > > **Regards ESS.** A key factor driving ESS is the quality of the proposal, which is shaped by guidance. In SURGE, even when stochasticity is present, the guidance pulls spread particles back toward the high-likelihood region. After guidance, most particles concentrate near the correct values, and their weights remain more uniform. This suggests that the observed ESS reflects the effectiveness of guidance in aligning particles with the posterior, rather than due to small observation noise. When the task is harder, SURGE naturally becomes more selective, discarding low-weight particles through resampling.
> > >
> > > To verify this, we ran an ablation on two high-dimensional tasks (Navier-Stokes and Weather, N=50) on a subset, replacing FlowDAS with a plain diffusion model that receives no observation guidance. ESS drops notably without guidance (NS: 20.9% → 7.7%; Weather: 51.7% → 19.7%), supporting the view that guidance is the driver of ESS.
> > >
> > > | Method/Metrics | Mean ESS/K |
> > > |------|-----|
> > > | DM Navier-stokes | 7.7% |
> > > | FlowDAS Navier-stokes | 20.9% |
> > > | DM Weather | 19.7% |
> > > | FlowDAS Weather | 51.7% |
> > >
> > > Thanks again for your time and feedback. We hope our response has clarified these points and would be grateful if you could reconsider the rating.

---

### Official Review · Reviewer_dnGm · 2026-03-11

**Soundness:** 3
**Presentation:** 3
**Significance:** 2
**Originality:** 3
**Overall Recommendation:** 4
**Confidence:** 4

**Summary:**

The paper is concerned with the data-assimilation problem where the dynamics is modeled by a diffusion generative model. The key contribution is the development of the SURGE (Sequential Unbiased Resampling via Girsanov Estimation) particle filtering framework. The proposed algorithm is similar to the sequential importance sampling particle filter, with the difference that the proposal density originates from a continuous-time diffusion model. Computation of the weights require computation of the density of the guided diffusion in comparison with the original one, which is achieved via Girsanov theorem. The method is evaluated on several benchmarks, including the Lorenz system, Navier–Stokes flow, and a weather forecasting dataset, where it demonstrates improved performance over baseline diffusion-based data assimilation methods.

**Compliance With Llm Reviewing Policy:**

Affirmed.

**Final Justification:**

I raised my score after reading the rebuttal and addressing the comparison to proposal methods in SIR. However, the concern about error analysis still remains, specially in the incremental likelihood setting.

**Key Questions For Authors:**

Is it possible to provide an error analysis of the proposed SIR approach?

**Limitations:**

The limitation discussion is not extensive.

**Strengths And Weaknesses:**

__Strength__: In my view, the main strengths of the paper lie in the clarity of the methodological description and the technical development. The proposed method is presented in a reasonably structured way, and the main ideas of the algorithm are explained clearly. The derivations and technical arguments appear largely correct to the best of my understanding, and the paper provides a coherent formulation of the filtering procedure and its implementation.

__Weakness__: The main limitations of the paper concern the originality and broader significance of the proposed approach.
-  __Relation to existing particle filtering literature__: The guided diffusion model can be interpreted as modifying the proposal distribution in a sequential importance resampling (SIR) particle filter in order to reduce the variance of the importance weights. Designing improved proposal distributions for particle filters is a well-studied topic. In this context, an important question is whether the proposed proposal density (Eq. (4)) has any optimality properties with respect to the variance of the importance weights. The paper does not analyze this point. For example, it would be useful to understand whether the choice of the function $G$  can be justified theoretically as reducing weight variance or improving sampling efficiency.
- __Incremental likelihood incorporation__: The idea of incorporating the likelihood gradually over time resembles approaches studied under homotopy filtering methods. While such strategies can improve weight stability, they also introduce additional resampling steps, which may increase the overall variance of the estimator. As the step size becomes small, the number of resampling steps grows, potentially amplifying resampling noise. Since the method is fundamentally based on the SIR framework, it would strengthen the paper to include an analysis of the resulting error behavior and its scalability, particularly with respect to the curse of dimensionality.

- __Role of the diffusion generative model__: The role of the generative diffusion model is not entirely clear. From the formulation, the method appears to operate on trajectories generated by an SDE-based dynamics (Eq. (3)). It would be helpful for the authors to clarify what conceptual or algorithmic difference arises when this dynamics is produced by a trained diffusion model rather than by a known dynamical system. In particular, it is not obvious what new challenges or advantages arise from the use of a learned diffusion surrogate in the filtering framework. Clarifying this point would help better position the contribution.

---

> ### Author Rebuttal · Authors · 2026-03-31
>
> We thank the reviewer for the careful feedback and are glad our method is clearly presented and derivations correct. We address each point below.
> On the relation to existing particle filtering literature.
> We agree that our guided diffusion can be interpreted through proposal design in sequential importance sampling. However, our key point is that guidance changes the full path-space proposal law induced by the guided diffusion, after which SURGE applies a Girsanov-based correction and resampling to recover the target posterior on trajectories, not only at the terminal state. SURGE is thus closer to a path-space SMC correction of an approximately guided diffusion than to a standard SIR filter with a hand-designed terminal proposal. This distinction is central and already reflected in our formulation based on path measures and Radon–Nikodym derivatives.
> We also agree that guidance's role should be more explicit theoretically. Guidance steers particles toward higher posterior mass, making the guided proposal closer to the posterior than the unguided prior. As in importance sampling generally, a closer proposal yields better-behaved importance weights with reduced variance. The purpose of guidance is not to claim exact optimality, but to produce a proposal empirically and conceptually closer to the posterior (quantified via Hellinger distance), improving sampling efficiency before Girsanov correction. We will clarify that our claim is variance reduction through posterior-directed guidance, rather than optimality of Eq. (4) in a strict global sense.
> The guidance term modifies the drift so the resulting proposal path measure approaches the posterior path measure, improving importance weight stability. We will revise the discussion to better position SURGE relative to improved proposal design in particle filtering. Importantly, this proposal-shaping mechanism is induced by the diffusion surrogate together with the guidance rule, rather than a conventional hand-crafted proposal; making this path-space correction precise for guided diffusion surrogates is part of our contribution.
> On incremental likelihood incorporation and error behavior.
> We appreciate this point. Progressive likelihood incorporation avoids a single highly concentrated weight update, keeping incremental weights close to one across small substeps, thus reducing resampling weight variability. Our formulation also suggests better guidance leads to smaller variance. In the special case where GG
> G solves a Bellman equation (simulable via Monte Carlo methods [1]), Itô's lemma implies the reweighting factor is exactly 1. More generally, the log reweighting factor magnitude can be bounded by the Bellman error. This analysis yields a scaling law balancing Monte Carlo cost of simulating Doob's h-transform, particle count, and generation steps. Rigorous formalization would require substantial additional analysis beyond this paper's scope. We will add a discussion in the revised version.
> Formalizing these observations rigorously would require a substantial amount of additional analysis, likely beyond the scope of the current paper. We therefore leave this direction for future work and will add a discussion of it in the revised version.
> Quantitatively, our intended error decomposition is as follows. The incremental likelihood is distributed over small substeps, **the corresponding incremental weights stay close to one as $\Delta t \to 0$, which implies that the variance introduced at each resampling step is also small, of order $O(\Delta t)$**. Summed over $T/\Delta t$ substeps over a fixed finite horizon, this yields an $O(1)$ total contribution rather than an exploding variance as $\Delta t \to 0$.  We agree that this argument should appear explicitly in the paper, and we will add a discussion clarifying the separate roles of discretization bias, resampling variance, and particle approximation error.
> At the same time, we also agree with the reviewer that this does not remove the curse of dimensionality in a worst-case sense. Our claim is more modest: the progressive likelihood incorporation and guided proposal improve weight stability and practical scalability relative to unguided or one-shot reweighting schemes, but they do not constitute a general dimension-free solution to particle degeneracy.
> Thanks again for your efforts into reviewing this paper. We hope we have addressed your concerns regarding the connection to particle filtering literature, the error analysis under discretization, and the role of the diffusion model. We are more than happy if you can reconsider the review rating.
>  [1] Zhu, Qijie, et al. “Training-Free Adaptation of Diffusion Models via Doob's $h$-Transform”. arXiv preprint arXiv:2602.16198, 2026.

---

> > ### Author Rebuttal · Reviewer_dnGm · 2026-04-03
> >
> > Thank you for your response. The "role of diffusion generative model" is still not clear. The question about error analysis is answered partially.

---

> > > ### Author Response · Authors · 2026-04-08
> > >
> > > We thank the reviewer for the updated recommendation and the constructive engagement throughout the review process.
> > >
> > > **Regards the role of diffusion generative model.** The key distinction is that in a diffusion surrogate, the drift function is a neural network, making each evaluation expensive (a full forward step of a deep neural network). Our approach uses guidance from the diffusion modeling literature to push particles toward the posterior region with minimal additional cost, which is specifically tailored to this computational constraint. We have revised the manuscript to make this distinction clearer.
> > >
> > > Thanks again for reviewing this paper and for raising the score. We appreciate the helpful discussion throughout the process.

---

### Decision · Program_Chairs · 2026-04-30

**Decision:**

Accept (regular)

**Comment:**

The paper studies data assimilation when system dynamics are generated by a pretrained diffusion surrogate. It treats the diffusion model as a digital twin and the observations as a physical twin that corrects the forecast over time. SURGE turns guided diffusion sampling into path-space importance sampling with Girsanov weights and resampling. It also adds progressive likelihood incorporation to keep weights stable during long rollouts. The paper evaluates the method on Lorenz, Navier-Stokes, and SEVIR weather forecasting.

The paper's main strength is that it gives a clear path-space formulation of diffusion-based filtering, which is the central technical contribution recognized across the reviews. The rebuttal also helped by clarifying the meaning of the weights and tightening the claim language, so the strongest overstatement was corrected rather than defended. Empirically, the gains are consistent across three difficult settings, including a real weather task, which makes the method feel like more than a toy proof of concept. Another practical advantage is that the method operates purely at inference time, so it can be applied to an existing diffusion surrogate without retraining.

The audience is researchers working on diffusion models, particle filters, and data assimilation, and they will read this paper for the path-space correction view and the practical inference-time recipe. The Girsanov-based correction of guided diffusion trajectories is a clean way to connect diffusion surrogates and sequential Monte Carlo. The empirical gains are consistent across three hard settings, and the paper is likely to be cited for that bridge between guided diffusion and filtering.